# ROOOH: A missing piece of the puzzle for OH measurements in low NO environments?

Christa Fittschen[1], Mohamad Al Ajami[1], Sebastien Batut[1], Valerio Ferracci[2,3], Scott Archer-Nicholls[2], Alexander T. Archibald[2,4], Coralie Schoemaecker[1]

[1]Université Lille, CNRS, UMR 8522, PhysicoChimie des Processus de Combustion et de l'Atmosphère - PC2A, Lille, 59000, France
[2]University of Cambridge, Centre for Atmospheric Science, Department of Chemistry, Lensfield Road, Cambridge, CB2 1EW, UK
[3]Cranfield University, Centre for Environmental and Agricultural Informatics, College Road, Cranfield MK43 0AL, UK
[4]National Centre for Atmospheric Science, Cambridge, UK

*Correspondence to*: Christa Fittschen (christa.fittschen@univ-lille1.fr)

**Abstract.** Field campaigns have been carried out with the FAGE technique in remote biogenic environments in the last decade to quantify the *in situ* concentrations of OH, the main oxidant in the atmosphere. These data have revealed concentrations of OH radicals up to a factor of 10 higher than predicted by models, whereby the disagreement increases with decreasing NO concentration. This was interpreted as a major lack in our understanding of the chemistry of biogenic VOCs, particularly isoprene, which are dominant in remote pristine conditions. But interferences in these measurements of unknown origin have also been discovered for some FAGE instruments: using a pre-injector, all ambient OH is removed by fast reaction before entering the FAGE cell, and any remaining OH signal can be attributed to an interference. This technique is now systematically used for FAGE measurements, allowing the reliable quantification of ambient OH concentrations along with the signal due to interference OH. However, the disagreement between modelled and measured high OH concentrations of earlier field campaigns as well as the origin of the now-quantifiable background-OH is still not understood. We present in this paper the compelling idea that this interference, and thus the disagreement between model and measurement in earlier field campaigns, might be at least partially due to the unexpected decomposition of a new class of molecule, ROOOH, within the FAGE instruments. This idea is based on experiments, obtained with the FAGE set-up of University Lille, and supported by a modelling study. Even though the occurrence of this interference will be highly dependent on the design and measurement conditions of different FAGE instruments, including ROOOH in atmospheric chemistry models might reflect a missing piece of the puzzle in our understanding of OH in clean atmospheres.

## 1 Introduction

OH radicals are the most important oxidant in the atmosphere, and the detailed understanding of their formation and reactivity is key for the understanding of the overall chemistry. Upon reaction with Volatile Organic Compounds (VOCs,

such as methane and isoprene), OH oxidation leads to the production of organic peroxy radicals ($RO_2$) who play a crucial role in the chemistry of tropospheric ozone and secondary organic aerosol (Monks et al., 2015). The concentration of OH radicals has been measured for several decades now (Holland et al., 2003;Creasey et al., 1997;Brune et al., 1995), and comparison of OH concentration profiles with model outputs is taken as a good indicator on the degree of understanding of the chemistry going on. Good agreement is often obtained between measurements and models for polluted environments (where levels of nitrogen oxides ($NO_x=NO+NO_2$) are in excess of 500 pmol/mol, or ppt), however remote and clean environments show much less good agreement (Stone et al., 2012). Several field campaigns in remote environments, dominated by natural biogenic emissions, have been carried out during the last decade (Whalley et al., 2011;Lelieveld et al., 2008;Hofzumahaus et al., 2009), and a very poor agreement has been found, with measured OH concentrations exceeding model predictions by up to a factor of 10. These findings have been interpreted to reflect a lack in our understanding of the oxidation mechanism of biogenic VOCs under low $NO_x$ conditions and have triggered a large number of studies aiming at improving the atmospheric oxidation mechanism of biogenic VOCs (Peeters et al., 2009;Crounse et al., 2011;Paulot et al., 2009;Archibald et al., 2010). Improvements have been made especially in the oxidation mechanism of isoprene (Wennberg et al., 2018), and new reaction pathways leading to OH recycling have been found. However, none of these new chemical pathways has led to a sufficiently significant increase in modelled OH concentration to bring models into reasonable agreement with measurements (Rohrer et al., 2014).

An alternative explanation for the unexpectedly high OH concentrations measured in biogenic, low NO environments is that the measurements suffer from an unidentified interference. Indeed, all of these measurements have been carried out using a technique named FAGE (Fluorescence Assay by Gas Expansion). Briefly, ambient air is rapidly expanded into a low pressure volume, where OH radicals are excited by 308 nm light and the resulting fluorescence is detected (Heard and Pilling, 2003). Calibration of the fluorescence signal allows the determination of absolute concentrations (Dusanter et al., 2008). Interferences can arise from different sources such as photolysis of suitable precursors by the fluorescence excitation laser (*e.g.*, $O_3$), the presence of fluorescing species other than OH or the decomposition of labile species during the gas expansion into the FAGE cell (Ren et al., 2004). The first source of interference can, in principle, be identified by varying the excitation laser energy: ambient OH radicals only need one photon to fluoresce, whilst other species need two (one for generating OH radicals by photolysis, another for their excitation). Therefore the fluorescence intensity would not vary linearly with the excitation energy. Even though in practice this method is highly uncertain, given the generally low OH concentrations (and the resulting low S/N ratio) and the high temporal variability of OH radical concentration, the high OH concentrations observed in the different field campaigns seems to arise from ambient OH and not from the photolysis of other species. This was also confirmed by Novelli et al (Novelli et al., 2014a) who observed a strong background during HUMPPA2010 with good S/N ratio, allowing to unequivocally exclude photolysis being at the origin of the background signal. The second source of interference can be identified by regularly measuring the fluorescence signal with the excitation laser wavelengths slightly tuned off the OH line. This procedure is always adopted during measurements as it enables to account for stray light reaching the detector from the excitation laser or from the sun.

The third source of interference, the generation of OH radicals during the expansion into the FAGE cell, is more difficult to identify because only one photon is needed and hence the interfering species would appear as ambient OH. Following the large disagreements between measurements and models, the group of W. Brune has redesigned a concept to quantify such possible interferences (Mao et al., 2012), that had first been tested by (Dubey et al., 1996): a pre-injector device is installed

just above the inlet into the FAGE cell, which injects regularly into the airflow a high concentration of a species rapidly reacting with OH radicals. This way all ambient OH radicals are scavenged before entering the FAGE cell, and any remaining signal can be identified as interference. The difference between the signal with and without the scavenger allows the quantification of the real ambient OH. The use of this technique was reported for the first time in 2012 showing results for a field campaign in a forest in California (Mao et al., 2012). It led to the identification of a large fluorescence signal

following scavenging of all ambient OH radicals, corresponding to up to 50% of the total OH concentration. The OH concentrations obtained with the scavenger agreed well with models, while the OH concentrations obtained without the scavenger exceeded modeled concentrations by up to a factor of 3. Other groups have also developed a pre-injector system in the following years (Griffith et al., 2016;Novelli et al., 2014a;Tan et al., 2017). Using this system, Novelli *et al*. (Novelli et al., 2014a) have observed strong interferences in their FAGE system during three field campaigns in remote biogenic

environments in Germany, Finland and Spain, while Griffith *et al*. (Griffith et al., 2016) was able to account for the observations through known interferences by $O_3$ photolysis. Tan *et al*. (Tan et al., 2017) have very recently observed a small unexplained OH concentration using a prototype pre-injector device during a field campaign in rural China. However, technical difficulties with the prototype made it uncertain to draw final conclusions about the origin of this unexplained OH signal.

Novelli *et al*. proposed that ozonolysis of alkenes, leading to the formation of Criegee intermediates and the subsequent decomposition of these Criegee intermediates within the FAGE cell, was responsible for the interference (Novelli et al., 2017). However, using different FAGE systems, Rickly and Stevens (Rickly and Stevens, 2018) and Fuchs *et al*. (Fuchs et al., 2016) could not confirm this source: even though they detected internally formed OH when mixing high concentrations of $O_3$ and alkenes in the laboratory, when they extrapolated their results to ambient conditions they found that the possible

interference generated this way would be well below the detection limit of their FAGE. Chamber studies were carried out at the SAPHIR chamber in Jülich (Fuchs et al., 2012), simulating remote forest conditions (*i.e.*, high biogenic VOC and low NO concentrations). OH concentrations were measured simultaneously by FAGE and by absolute DOAS absorption. No sizeable interference was detected in these experiments, even though the same group had previously observed unexpected high OH concentrations in the Pearl River delta in China (Hofzumahaus et al., 2009;Rohrer et al., 2014), exceeding modelled

concentrations by up to a factor of 8.

Following several years of interference studies in various environments, recent work from W. Brune's group (Feiner et al., 2016) concluded that the interference observed in their FAGE system (a) was due to a rather long-lived species because the interference persists into the evening, (b) it had been observed in different environments dominated by MBO, terpenes or isoprene, hence it must originate from a class of species rather than from only one species such as isoprene, (c) it strongly

increased with increasing O($^1$D), hence it must somehow be linked to photochemistry and (d) the species responsible for this interference was linked to a low NO$_x$ oxidation pathway, because the extent of the interference steeply decreased with increasing NO concentration.

In this work we present experimental and modelling evidence that this sought-after species could be the product of the reaction between RO$_2$ radicals and OH radicals. In recent works it has been shown that this reaction is fast (Assaf et al., 2017b;Assaf et al., 2016;Yan et al., 2016) and could be competitive to other sinks for RO$_2$ radicals (Fittschen et al., 2014;Archibald et al., 2009), *i.e.* it becomes increasingly important with decreasing NO concentration. *Ab-initio* calculations (Müller et al., 2016;Liu et al., 2017;Assaf et al., 2018a) have shown that the initial reaction product is a trioxide, ROOOH, obtained from the recombination of RO$_2$ and OH. The formation of this adduct is exothermic by around 120 kJ mol$^{-1}$ compared to the initial reaction partners and by around 110 kJ mol$^{-1}$ compared to the major decomposition products, RO + HO$_2$, largely independent of the size of the alkyl moiety of the RO$_2$. While for the smallest RO$_2$ radical, CH$_3$O$_2$, stabilization of CH$_3$OOOH is not the major fate of the initial adduct (Assaf et al., 2017a;Müller et al., 2016;Caravan et al., 2018) and the major products are CH$_3$O + HO$_2$, the HO$_2$ yield has been found to decrease with increasing size of the alkyl group and it is expected that for C$_4$ peroxy radicals the stabilization of the initially formed ROOOH is the major product (Assaf et al., 2018b). For RO$_2$ radicals obtained from an initial attack of OH radicals on biogenic VOCs, it can thus be expected that the major reaction product of the reaction between these RO$_2$ radicals with OH radicals will also be the corresponding trioxides. Depending on the removal rate of ROOOH (which is not known to date), sizeable concentrations of this new class of species can possibly accumulate and thus be present in low NO environments.

## 2. Results and Discussion

In the first part, the experimental evidence for the interference generated in the UL-FAGE by the presence of ROOOH molecules will be presented. It should be noted that the intensity of interferences or even the presence at all can depend on the design of the FAGE instrument (inlet design, pressure drop, residence time etc) and the results presented here are only valid for the FAGE instrument of University Lille. Other FAGE instruments need to be tested individually for the possible presence of an interference in OH measurement due to the presence of ROOOH. In the second part, model calculations are used in order to estimate the steady state concentration of ROOOH molecules that can possibly build up in different environments.

### 2.1 Experiments

With the goal of forming sizeable amounts of trioxide (ROOOH), experiments have been carried out in a pump-probe FAGE instrument of the University Lille (UL-FAGE), describes already in detail in earlier publications (Fuchs et al., 2017;Hansen et al., 2015;Parker et al., 2011). Briefly, a gas mixture containing the VOC (isoprene,C$_4$H$_{10}$ or CH$_4$) and O$_3$/H$_2$O is photolysed at 266 nm at a repetition rate of 2 Hz. The photolysed mixture is expanded into the FAGE cell, and the OH concentration is monitored by time-resolved Laser Induced Fluorescence (LIF). The excitation laser operates at 5 kHz, hence

the OH profiles are obtained with a time resolution of 200 μs. The residence time of the gas mixture in the photolysis cell is around 20 s, therefore the mixture is photolysed around 40 times before it reaches the FAGE inlet. A schematic view of the experimental set-up is shown in **Figure 1**, more details can be found in the Supplementary data.

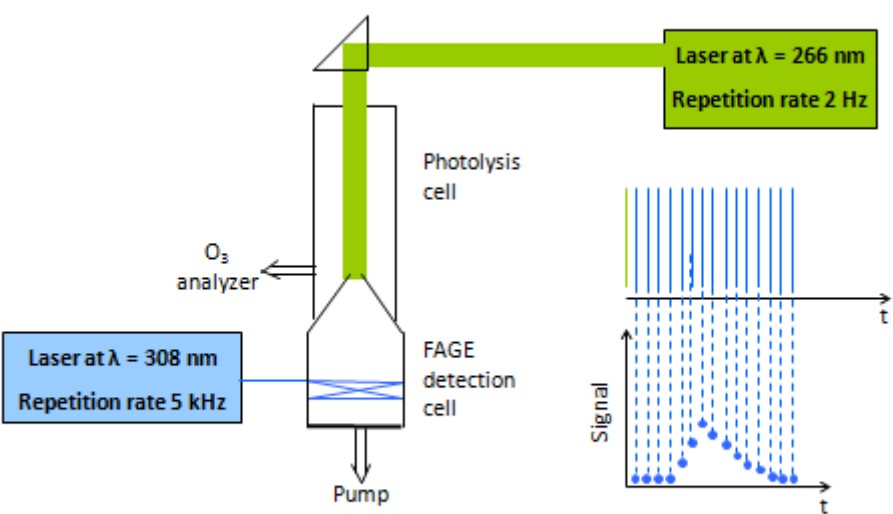

5 **Figure 1: Schematic view of the experimental set-up**

Experiments start with a fresh mixture (*i.e.*, with the photolysis laser manually covered) and 40 decays are then recorded every 0.5 s for 20 s. After 40 photolysis pulses the laser is covered again for 2 minutes to allow the mixture to completely refresh and, in order to improve the S/N ratio, a new series of measurements is started. After 20 series, the signals are
10   averaged so that one OH decay profile is obtained for each sequential photolysis pulse. An example is shown in **Figure 2** where, for clarity, only one in every 10th decay profile is plotted. The open black signals in **Figure 2** show the pre-photolysis signal, i.e. the signal registered just before uncovering the photolysis laser. This signal is not zero, because some stray light from the excitation laser is always detected. Also, some ambient laboratory light can reach the detector through the photolysis window and the nozzle opening.

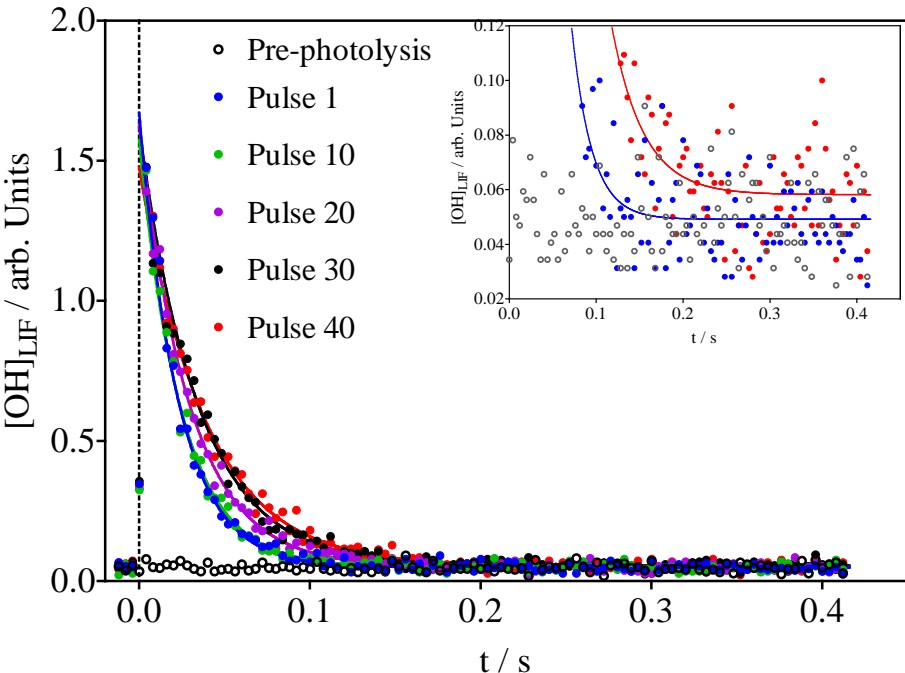

**Figure 2: OH concentration time profiles following the photolysis of 600 ppb $O_3$ (leading to initial OH concentrations of around $1.4 \times 10^{10}$ cm$^{-3}$) in the presence of $3 \times 10^{11}$ cm$^{-3}$ isoprene. For clarity, only every 10$^{th}$ photolysis shot is shown. Open black symbols show the FAGE signal before the first photolysis shot. Time resolution was decreased from 200 µs to 4 ms by averaging 20 data points for clarity only: full lines show a fit to a single exponential decay, carried out using non-averaged data between 0.02 s and the end of the data set. The inset shows a vertical zoom, for clarity only the pre-photolysis signal as well as the signals for the first and last pulse with the corresponding fits are shown.**

The initial isoprene concentration ($3 \times 10^{11}$ cm$^{-3}$ in **Figure 2**) was chosen to be low enough to make the reaction of $RO_2$ with OH compete efficiently with that of isoprene with OH after several photolysis pulses: with initial OH concentrations of $1.4 \times 10^{10}$ cm$^{-3}$ (obtained from calibration in separate experiments, see Supplementary data), the isoprene concentration decreases with each photolysis shot, while the $RO_2$ radical concentration increases. It can thus be expected that the concentration of ROOOH increases with every photolysis pulse. With the goal of getting a good idea of the ongoing chemistry in the photolysis cell and to get a rough estimate of the concentration of ROOOH being produced during this experiment, a very simple model was run using the conditions shown in **Figure 2**.

**Table 1: Model used to estimate the accumulation of ROOOH in the photolysis cell before entering the FAGE cell, all rate constants have been taken from the most recent IUPAC evaluations (Atkinson et al. 2006, Atkinson et al., 2004)**

| Reaction | k / cm$^3$ s$^{-1}$ |
|---|---|
|  |  |

| | |
|---|---|
| OH + Isoprene → $RO_2$ | $1 \times 10^{-10}$ |
| OH + $RO_2$ → ROOOH | $1 \times 10^{-10}$ |
| OH + ROOOH → products | $1 \times 10^{-11}$ |
| OH + $O_3$ → $HO_2$ + $O_2$ | $7.3 \times 10^{-14}$ |
| OH + $HO_2$ → $H_2O$ + $O_2$ | $1 \times 10^{-10}$ |
| $RO_2$ + $RO_2$ → products | $1 \times 10^{-12}$ |
| $RO_2$ + $HO_2$ → ROOH | $1.7 \times 10^{-11}$ |

A yield of 1 is estimated for the formation of ROOOH in the reaction of $RO_2$ with OH. The other major reaction path for the $RO_2$ radicals under these conditions is the self-reaction. The reaction of ROOOH with OH radicals has been estimated (in comparison with ROOH) to $1\times10^{-11}$ $cm^3s^{-1}$, but only a small fraction of ROOOH will have reacted with OH after 40 photolysis pulses.

This model was run 40 times for 0.5 s, with the final concentrations of the different species obtained at each run being used as initial concentrations in the following run, always adding $1.4\times10^{10}$ $cm^{-3}$ OH radicals to the mixture. The evolution of the different species with each photolysis shot is shown in **Figure 3**.

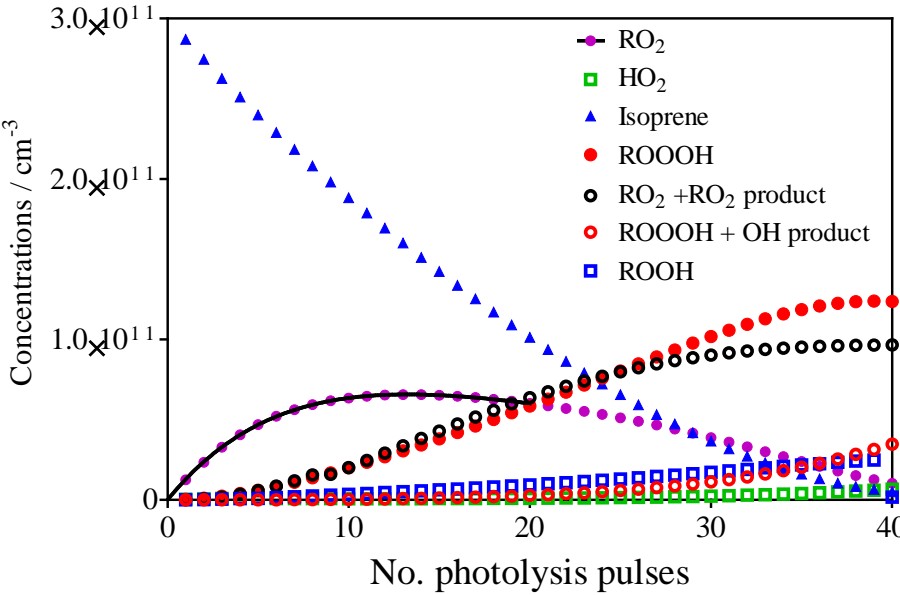

**Figure 3**: **Evolution of different species in the photolysis cell as a function of the number of photolysis pulses. Full black line describes evolution of $RO_2$ by exponential rise (see section on $CH_4$ experiments)**

The goal of this model is not to precisely describe the ongoing chemistry, but rather to get a good idea of how much ROOOH is possibly accumulated. The model uses different simplifications: (i) OH radicals only react with species present in the model, i.e. no wall loss or reaction with impurities is taken into account; (ii) the possible photolysis of ROOOH at 266 nm or a heterogeneous loss on the reactor walls is not taken into account; (iii) no reaction of OH with the products of $RO_2$ self-reaction are considered; (iv) the photolysis beam has been considered homogeneous, the inhomogeneity of the beam profile of our photolysis laser has not been considered., (v) the decrease of available $RO_2$ concentration due to diffusion of the photolysed volume with fresh gas mixture from outside the photolysis beam. All these simplifications lead to an uncertainty in the final ROOOH concentration, possibly up to a factor of 10. Most of the simplifications will lead to an overestimation of the final ROOOH concentration (either ROOOH is consumed or less is formed), except the inhomogeneous photolysis beam where the direction of uncertainty is not easy to determine (higher formation of ROOOH in the hotspots of the laser beam and lower in the rest of the volume). The model predicts the formation of around [ROOOH] $\approx$ $1\times10^{11}$ cm$^{-3}$.

The model predicts the consumption of most isoprene over the 40 photolysis pulses, which should lead to a decrease in the decay rate, given the much faster rate constant of OH with isoprene compared to the reaction products. A single-exponential decay was then fitted to the experimental OH profiles from **Figure 2** and the resulting pseudo-first order decay rates are shown as blue dots in **Figure 4**. It can be seen that the decay rate decreases over the 40 shots by around 20 s$^{-1}$ , corresponding to a decrease in isoprene concentration of around $2\times10^{11}$ cm$^{-3}$, in good agreement with predictions of a kinetic model. The OH LIF signal at long reaction times, obtained as the plateau of the single-exponential fit (red dots in **Figure 4**), increases with increasing number of photolysis pulses (m = (1.2±0.3) × 10$^{-4}$ arb. Units / photolysis pulse). This can be interpreted as interference due to decomposition of the increased concentration of ROOOH within the FAGE, however more tests will be presented further down to strengthen this hypothesis.

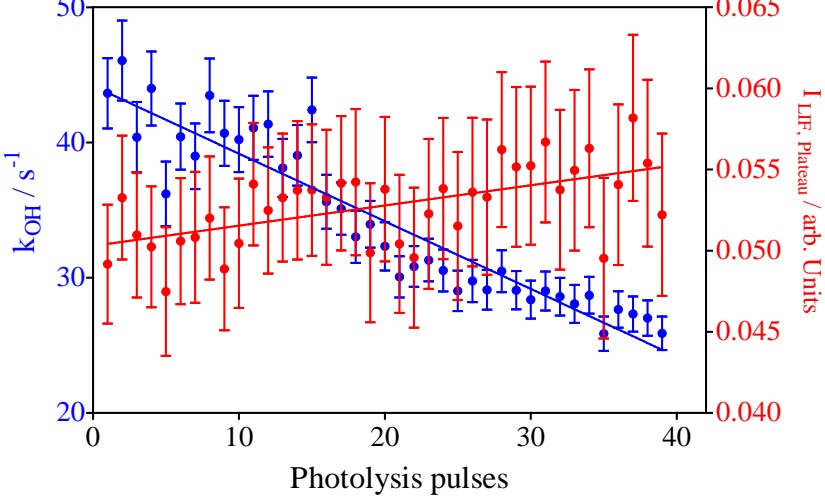

**Figure 4: Results of fitting a mono-exponential decay to the raw signal of the experiments shown in Figure 1. Blue dots: OH decay rates from the mono-exponential fit between 0.02 s and the end of the data set (left *y*-axis). Red dots: fluorescence signal after reaction of all OH radicals obtained as plateau of the single-exponential fit (right *y*-axis). Error bars show 95% confidence interval such as obtained from the fit.**

In order to better understand the origin of the increase of the LIF signal, additional experiments have been carried out.

### 2.1.1. Is the increase due to a 1- or 2-photon process?

Photolytically generated interferences need two photons for generating one fluorescence photon, and can thus be identified

10    by either varying the fluorescence excitation laser energy (the signal intensity would increase with the square of the excitation laser energy) or by changing the repetition rate of the excitation laser (photolytically generated interferences appear because the air mass within the excitation volume is not completely renewed between two excitation laser pulses (200µs at 5 kHz), and thus OH radicals generated with one pulse can be excited with the following pulse. Hence, such interference would be expected to decrease with decreasing repetition rate). Separate test experiments with $CH_3COCH_3$ as a

15    known source of photolytically generated OH radicals are described in the supplementary data. Three experiments with isoprene ($3.2 \times 10^{11}$ cm$^{-3}$) have been carried out, keeping all other parameters constant (266nm photolysis energy and repetition rate, $O_3$ concentration): two experiments at 5 kHz with different excitation laser energies (1.7 and 0.8 mW) and one series with a lower excitation laser repetition rate (1 kHz, 0.4 mW). The results are shown in **Figure 5**.

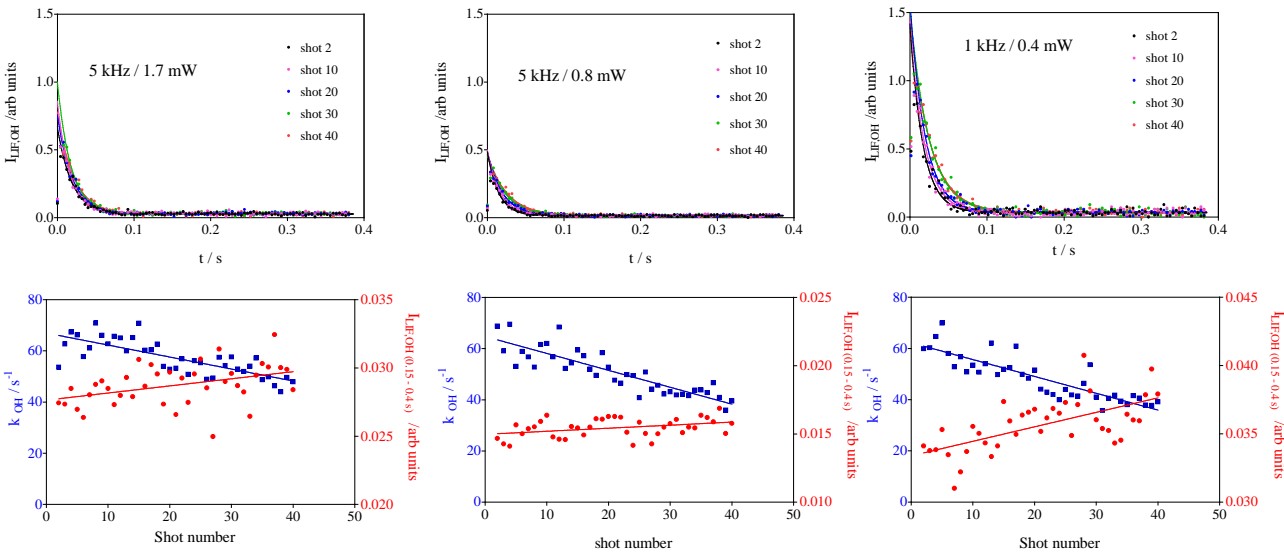

**Figure 5: Photolysis of $O_3$ in the presence of isoprene using different excitation laser energies and repetition rates. Upper graphs: OH decays (for clarity, only every 10$^{th}$ decay is shown), lower graph OH decay rate as a function of photolysis pulses (blue dots, left y-axis) and fluorescence intensity averaged over 0.15 to 0.4 s (red dots, right y-axis).**

The blue dots on the lower graphs show the decrease in the decay rate with increasing number of photolysis pulses, on the same order of magnitude for all three experiments, as expected (photolysis energies as well as isoprene and $O_3$ concentration were identical for all three experiments). The absolute values for the background signals are different for the three experiments, they are highest for the highest pulse energy (0.4 mW at 1 kHz) and lowest at the lowest pulse energy (0.8 mW at 5 kHz), reflecting that the laser stray light is partially at the origin of the "background background". The background increases with increasing number of photolysis pulses for all three series, but the slope is different. However, the slope is directly proportional to the sensitivity of the LIF detection system, and for comparison needs to be normalized to the initial OH intensity. The results are summarized in **Table 2**:

**Table 2**: Summary of results from **Figure 5**

| Experiment | $OH_0$ LIF intensity [a] | Slope [b] | Slope / $OH_0$ |
|---|---|---|---|
| 5 kHz, 1.7 mW | $0.85 \pm 0.08$ | $(5.2\pm2.0) \times 10^{-5}$ | $(6.1 \pm 2.5) \times 10^{-5}$ |
| 5 kHz, 0.8 mW | $0.48 \pm 0.04$ | $(2.2\pm0.9) \times 10^{-5}$ | $(4.6 \pm 2.3) \times 10^{-5}$ |
| 1 kHz, 0.4 mW | $1.50 \pm 0.17$ | $(10.0\pm3.1) \times 10^{-5}$ | $(6.7 \pm 2.7) \times 10^{-5}$ |

[a] **$OH_0$ LIF intensity obtained as the average of the LIF intensity at t=0 for all 40 photolysis pulses, obtained by fitting to a single exponential decay between 0.015 – 0.4 s, in arbitrary units**
[b]. **Slope obtained by linear regression of red dots in Figure 5, in arbitrary units**

From the observation that the increase in residual LIF signal with increasing number of photolysis pulses is independent of both (a) the fluorescence excitation laser energy and (b) the repetition rate of the excitation laser, we conclude that the observed interference in the UL-FAGE is not due to a photolytic process.

## 2.1.2. Is the interference really due to the product of $RO_2$ + OH?

Additional experiments have been carried out using identical OH concentrations, but much higher isoprene concentrations than in the above experiments. Under these conditions, there is still formation of high concentrations of $RO_2$, but as the isoprene concentration stays high even after 40 photolysis pulses, the $RO_2$ concentration never gets high enough to compete with the reaction of isoprene with OH. Therefore, one can expect comparable formation of all products from $RO_2$ self- or cross reaction or reaction with $HO_2$, but only very little or no products from the reaction of $RO_2$ with OH.

The results of these experiments are shown in **Figure 6**. For the conditions in the left graph ($[C_5H_8] = 1.23 \times 10^{12}$ cm$^{-3}$) the OH decay rate decreases ((-0.5$\pm$0.2) s$^{-1}$ pulse$^{-1}$ = 20 s$^{-1}$ after 40 pulses) in the same way than for the experiments above, and this is explained by the replacement of the reactive isoprene by less reactive products. For the conditions in the right graph the $C_5H_8$ concentration was so high ($[C_5H_8] = 1.23 \times 10^{13}$ cm$^{-3}$) that it leads to decay rates that are not measurable anymore

with our time resolution. For both conditions however, the LIF-intensity at long times does not increase within the experimental uncertainty with the number of laser pulses ($(1.2\pm1.4) \times 10^{-5}$ and $(-1.3\pm1.2) \times 10^{-5}$ for the left and right graph, respectively).

From these observations, it can be concluded that the increase in LIF intensity at long reaction times observed in the experiments presented in Figure 4 is consistent with being generated by the product of the reaction between $RO_2$ radicals and OH radicals.

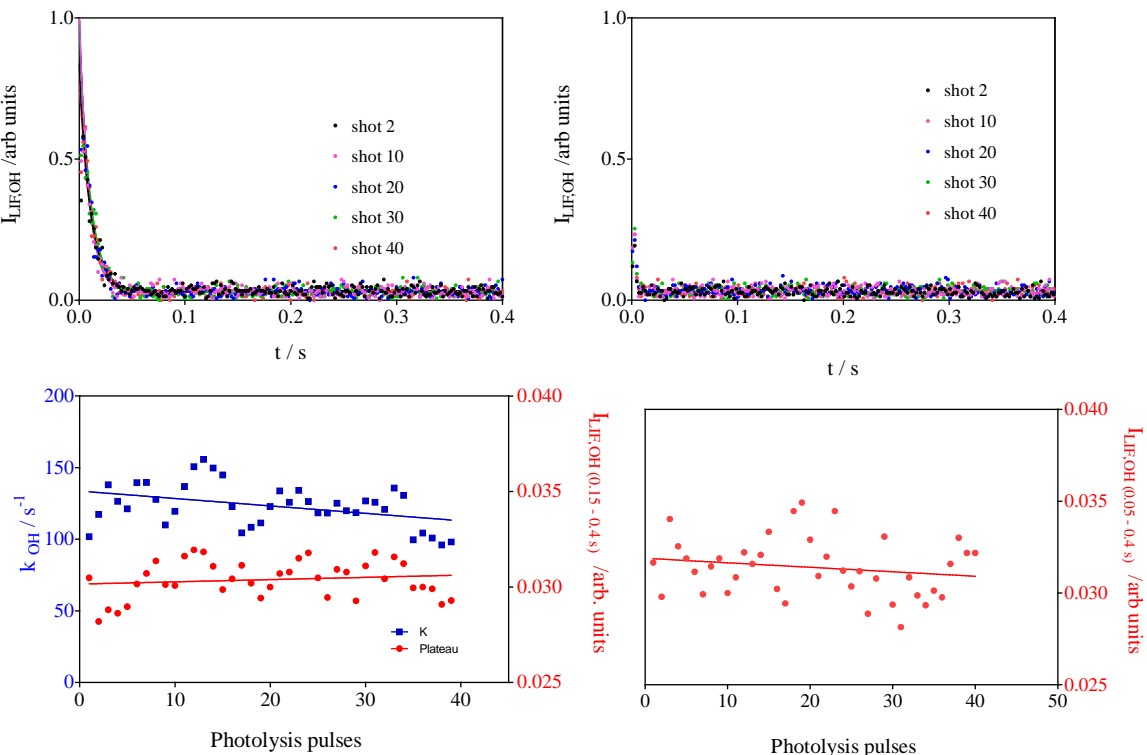

**Figure 6: Experiments with high isoprene concentrations: $[C_5H_8] = 1.23 \times 10^{12}$ and $1.23 \times 10^{13}$ molecule.cm$^{-3}$ for left and right graph, respectively. Upper graph LIF signals as a function of the number of photolysis pulses (for clarity, only every 10$^{th}$ pulse is shown), lower graph shows the rate constant in blue (left graph only, decay was too fast to be measurable under the conditions of the right graph) and the LIF intensity at long times (plateau from fitting for left graph, average of all data points between 0.01 – 0.4 s for right graph).**

### 2.1.3. Tests with n-C$_4$H$_{10}$

To further support the hypothesis that the observed increase in residual LIF signal is due to an interference generated by the product of the reaction of $RO_2$ + OH, additional experiments have been carried out with $C_4H_{10}$ instead of isoprene. $C_4H_{10}$ has been chosen because Assaf et al. (Assaf et al., 2018b) have shown experimentally that the HO$_2$ yield for the reaction of the

corresponding $RO_2$ radical with OH is very low and *ab-initio* and RRKM calculations support the hypothesis that the major reaction product with increasing alkyl size of the $RO_2$ radical becomes the corresponding trioxide. For the reaction of the corresponding isoprene peroxy radical with OH such direct evidence is currently not available, and it could be imagined that the OH radicals would rather add to the remaining double bond rather than to the peroxy site. Note however, that the major conclusion from the above experiments (the product of the reaction between the isoprene-peroxy radical with OH generates an interference in the UL-FAGE) would still be the same. Three experiments with different butane concentrations have been carried out and the results are shown in **Figure 7**.

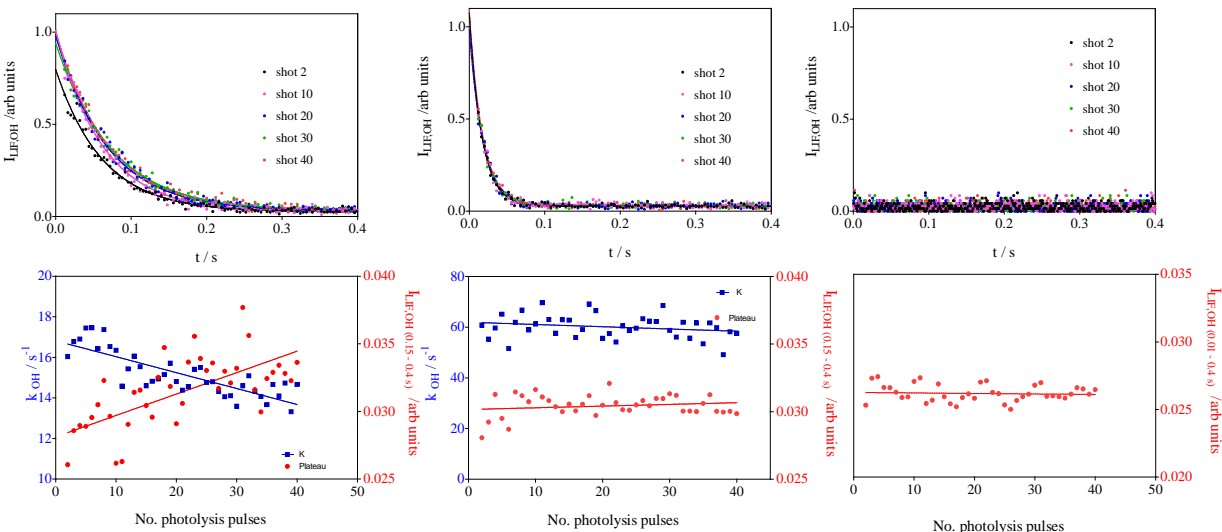

**Figure 7: Photolysis of $O_3$ in the presence different concentrations of n-butane ($7 \times 10^{12}$, $2 \times 10^{13}$ and $7.5 \times 10^{15}$ cm$^{-3}$ from left to right). Upper graph: OH decays (for clarity only every 10$^{th}$ decay is shown), lower graph: decay rates of OH radicals as a function of photolysis pulses (blue dots, left y-axis), residual LIF intensity taken from mono exponential fit for left graph and as the average LIF intensity between 0.15 – 0.4 s and 0.01 and 0.4 s for the center and right graph, respectively.**

For the lowest concentration (left graphs in **Figure 7**) a high formation of ROOOH can be expected: under these conditions OH radicals react slowly with butane and the reaction with the nascent $RO_2$ radicals becomes rapidly competitive. The concentration has been increased in the middle graph of **Figure 7** such that only a very low concentration of ROOOH is expected. In the right graph, finally, a very high concentration of butane has been used, too high to detect the decay of OH radicals with our time resolution. Under these conditions, it is expected that OH radicals react nearly exclusively with butane and no ROOOH is formed. Note that the initial OH radical concentration is the same in all three experiments. The interference is clearly visible in the left graph (slope $m = (15.8\pm4)\times10^{-5}$ arb. units), barely in the center graph ($m = (1.2\pm1.7)\times10^{-5}$ arb. units) and not present anymore in the right graph ($m = -(0.4\pm1.3)\times10^{-5}$ arb. units). Note that in the experiment of the right graph, the concentrations of all other species are similar to the concentrations in the left graph, i.e. the $RO_2$ and $HO_2$ concentrations are similar and with this all products obtained from self-and cross reactions. This is another

strong indicator that the observed increase in residual LIF intensity is indeed due to the product of the reaction of $RO_2$ with OH.

### 2.1.4 Tests with $CH_4$

5  Experiments with $CH_4$ have been carried out because it is known that the $HO_2$ yield in the reaction of $CH_3O_2$ with OH is very high, and that the yield of stabilized $CH_3OOOH$ is expected to be very low (Assaf et al., 2018b;Assaf et al., 2017a). Therefore, it would not be expected to observe an interference in the FAGE system. Two experiments with different $CH_4$ concentrations have been performed, the results are shown in **Figure 8**. In both series, one observes for the OH decay rate an increase over the first few photolysis shots. This is expected due to the formation of $CH_3O_2$ radicals that are more reactive

10  against OH radicals than $CH_4$. In **Figure 3** it can be seen that the model predicts (for an overall reactivity of 30 s$^{-1}$) an increase of $RO_2$ radicals over the first 10 pulses, followed by a steady state period and a slow decay. The decay rates are plotted as a function of the photolysis pulses in **Figure 8** (lower graphs) and have been fitted by forcing to the same rise time as the one obtained from the mono exponential fit of the $RO_2$ profile in **Figure 3**. A rough estimation of the increase in the decay rate of 8 s$^{-1}$ is obtained, corresponding to a $CH_3O_2$ concentration (using k($CH_3O_2$+OH) = 1.5×10$^{-10}$ cm$^3$s$^{-1}$) (Assaf et

15  al., 2016) of 5×10$^{10}$ cm$^{-3}$, in excellent agreement with the predictions of the model, **Figure 3**. This good agreement gives more confidence in the principle idea of the experiments and the conditions chosen to enhance the formation of ROOOH.

In both experiments the LIF intensity at long times does not change ((-3.0±2.5×10$^{-5}$ and 1.0±1.7×10$^{-5}$ for left and right graph, respectively). This is expected due to the small yield of $CH_3OOOH$ and further supports the hypothesis that ROOOH, the product of the reaction between $RO_2$ and OH, leads to an interference in UL-FAGE.

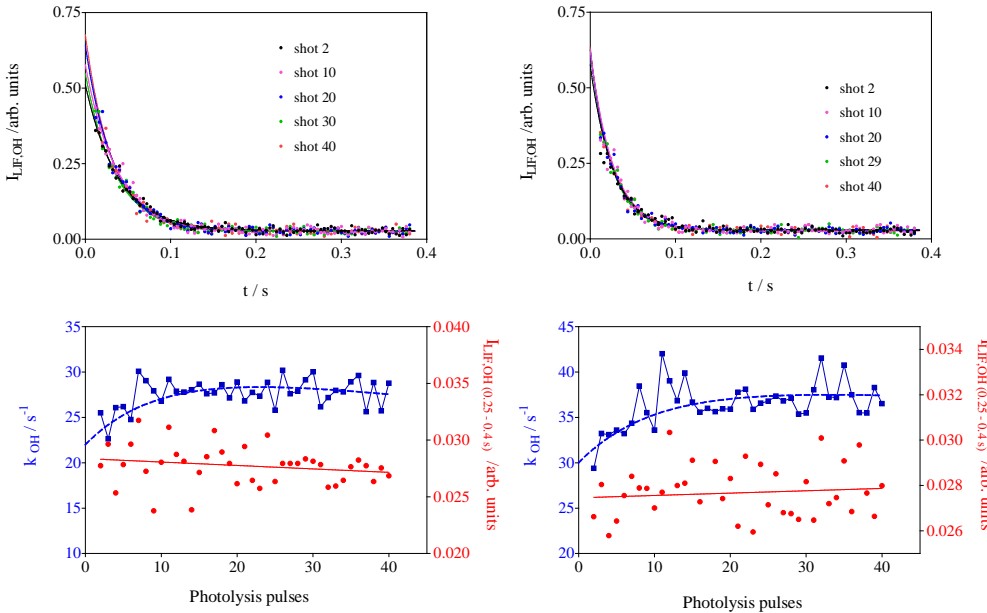

**Figure 8: Photolysis of $O_3$ in the presence different concentrations of $CH_4$ ($3.3 \times 10^{15}$ cm$^{-3}$ and $4.9 \times 10^{15}$ cm$^{-3}$ for the left and right graph, respectively). Upper graph: OH decays (for clarity only every 10$^{th}$ decay is shown), lower graph: decay rates of OH radicals as a function of photolysis pulses (blue dots, left y-axis), residual LIF intensity taken as the average LIF intensity between 0.25 – 0.4s.**

### 2.1.5 Intensity of interference in UL-FAGE

The increase in residual LIF signal in **Figure 4** over the 40 photolysis pulses is around 0.005 arb. units. This can be compared with the raw OH decays shown in **Figure 2**: the initial LIF signal of $\approx$ 1.7 arb. units corresponds to an OH concentration of $1.4 \times 10^{10}$ cm$^{-3}$. Therefore, the increase in the residual signal corresponds to an equivalent OH concentration

of $\approx 4 \times 10^7$ cm$^{-3}$. The concentration of ROOOH accumulated after 40 photolysis pulses was estimated to be [ROOOH] $\approx$ $1 \times 10^{11}$ cm$^{-3}$ using a simple model, i.e. a fraction of $\approx 4 \times 10^{-4}$ of ROOOH decomposed to OH radicals during the expansion within the UL-FAGE. No clear explanation can be given on the mechanism of this OH formation: a homogeneous decomposition within the shock wave of the expansion is unlikely, because the pathway leading to $CH_3O$ and $HO_2$ is thermodynamically more favoured (Assaf et al., 2018a) and thus no OH would be expected. Therefore a heterogeneous

decomposition on the walls of the FAGE cell or the entrance nozzle are more likely. The residence time of the gas mixture between entrance nozzle and detection beam can be calculated from the volume of the cell (0.25 l) and the gas flow (3 l min$^{-1}$ STP) to around 1 sec, leaving ample time for collisions with the reactor walls.

It can hence be concluded that in the UL-FAGE an interference signal corresponding to [OH] = $1 \times 10^6$ cm$^{-3}$ (order of magnitude of the disagreement between model and measurements) could be generated by less than 100 ppt of ROOOH. It

needs to be emphasized, that this result is only valid for the UL-FAGE and the magnitude or even presence of this interference might be very different for other FAGE instruments and needs to be tested. Also, it is not very likely that this species can explain the observed increase in the interference at night, such as observed by (Novelli et al., 2014a).

In order to estimate if ROOOH concentrations in this range can possibly be accumulated in remote biogenic environments, calculations using global and box models have been performed.


### 2.2 Modeling Results

The global distribution of ROOOH species produced by the $RO_2$ + OH reaction was investigated using the Met Office's Unified Model with the United Kingdom Chemistry and Aerosols scheme (UM-UKCA), version 8.4 (Abraham et al., 2012). UM-UKCA is a global chemistry-climate model with a horizontal resolution of 1.875° in longitude $\times$ 1.25° in latitude on 85

vertical levels from the surface up to a height of 85 km (in its N96-L85 configuration). The chemistry scheme and emissions used in the present study were described in detail in a recent work (Ferracci et al., 2018) and included isoprene oxidation (Archibald et al., 2010) and isoprene emissions.

Crucially, the model simulated the abundances of a number of peroxy radicals resulting from the oxidation of emitted VOCs: $CH_3O_2$ (methyl peroxy), $CH_3CH_2O_2$ (ethyl peroxy), $CH_3CH_2CH_2O_2$ (n-propyl peroxy), $(CH_3)_2CHO_2$ (i-propyl peroxy), $CH_3C(O)O_2$ (acetyl peroxy), $CH_3CH_2C(O)O_2$ (propionyl peroxy), $CH_3C(O)CH_2O_2$ (propyldioxy peroxy). Peroxy radicals from the first oxidation of isoprene were lumped into one species, as those from the oxidation of isoprene oxidation products (methacrolein and methyl vinyl ketone). These were used, along with the modelled number densities of OH and a rate constant $k_1$ of $1.5 \times 10^{-10}$ cm$^3$ s$^{-1}$ for all $RO_2$ + OH reactions (consistent with laboratory studies (Faragó et al., 2015;Assaf et al., 2017b;Assaf et al., 2016)) to calculate the total rate of production of ROOOH species. The total atmospheric abundance of trioxide species, [ROOOH]$_{ss}$, was then calculated offline assuming steady state between the production and loss ($L$) processes of ROOOH, according to the equation:

$$[ROOOH]_{SS} = \frac{k_1[OH]\sum_{i=1}^{n}[RO_{2,i}]}{L} \tag{1}$$

where the sum is across all $RO_2$ radicals in the model excluding methyl peroxy radicals, for which it has been shown that the production of a trioxide species is only a minor product channel (Assaf et al., 2017a) while the trioxide yield is expected to be close to 1 for larger peroxy radicals (Assaf et al., 2018b).

Steady-state ROOOH abundances were calculated "offline" using the modelled abundances of hourly [OH] and [$RO_2$] along with a rate constant (Assaf et al., 2016;Assaf et al., 2017b) for ROOOH formation of $1.5 \times 10^{-10}$ cm$^3$ s$^{-1}$. As neither the removal rate nor the dominant loss process of these ROOOH species are currently known, different removal rates were tested, ranging from $10^{-5}$ to $10^{-2}$ s$^{-1}$. In any case, the modelled [ROOOH] followed a diurnal and seasonal cycle similar to that of its precursors (OH and $RO_2$). Therefore the highest [ROOOH] values were found around midday-2pm in the summer months (JJA in the Northern Hemisphere, DJF in the Southern Hemisphere). The peak [ROOOH] values shown in **Figure 9** and in **Figure S4** were determined by producing an average seasonal diurnal cycle for each model grid cell and then plotting only its peak [ROOOH] value. **Figure 9** shows the average diurnal peak concentration of ROOOH in the Boreal (left) and Austral (right) summer obtained using a removal rate of $10^{-4}$ s$^{-1}$, leading to ROOOH lifetimes of around 3 hours, on the same order as the lifetime of ROOH species. Peak concentrations of several 100 ppt are reached in this scenario, especially at tropical latitudes, which would lead to an interference in the UL-FAGE system of the order of $1 \times 10^6$ cm$^{-3}$. However, we would like to emphasize that both, the modelled concentration of ROOOH in the atmosphere as well as the sensitivity of the UL-FAGE against ROOOH species, bear currently an uncertainty of at least a factor of 10.

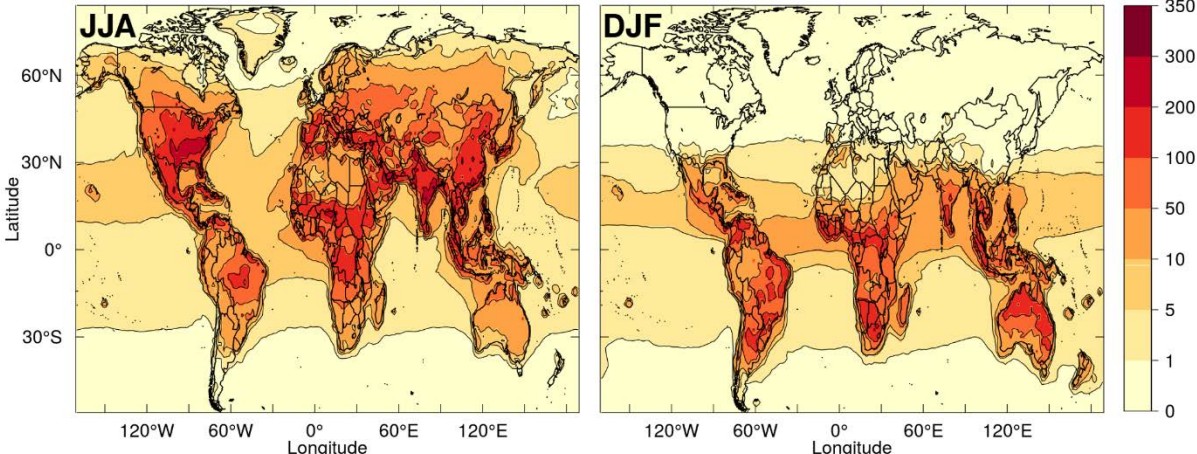

**Figure 9: Modelled mean diurnal peak ROOOH volume mixing ratio (in ppt) during Northern (left) and Southern (right) summer months, using a combined removal rate for all ROOOH of $10^{-4}\,\mathrm{s}^{-1}$.**

To confirm these global model results, a steady-state box model, constrained to observations (including OH, NO, Isoprene and HO$_2$) made in the South East USA (Feiner et al., 2016), was developed. The results of the calculations with the steady-state model are shown in **Figure 10**, which highlights that at low levels of [NO] (< 200 ppt), typical in remote BVOC rich environments, levels of [ROOOH] are predicted to be on the order of 50-200 ppt, with a steep increase at [NO] < 100 ppt. The two datasets plotted in **Figure 10** span a range of different NMVOC (isoprene) mixing ratios and highlight that ROOOH levels increase with increasing [VOC] and decreasing [NO], in agreement with the global 3D modelling results shown in **Figure 9**.

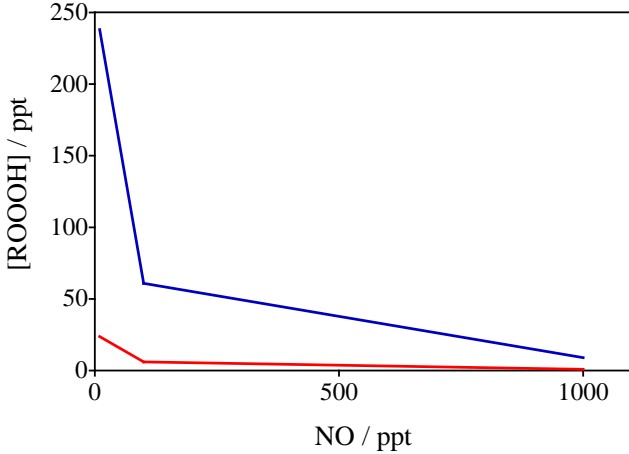

**Figure 10: Variation in ROOOH as a function of NO (x-axis) and VOC reactivity (different colours) constrained by data from Feiner et al. (Feiner et al., 2016). Those data in red reflect a situation of VOC reactivity of 5 s$^{-1}$ whilst the blue data reflect VOC reactivity of 24 s$^{-1}$ (similar to that seen in regions like the Amazon).**

## 3. Discussion

In this work we have shown that the product of the reaction of $RO_2$ radicals with OH radicals leads to an OH interference signal in the UL-FAGE instrument. The intensity of such interference or even the occurrence at all can depend on the design and working conditions of the FAGE set-up, which is different for different groups. However, if occurring also with a comparable intensity in other FAGE instruments, this interference might be high enough to explain numerous observations obtained with FAGE instruments from other research groups including:

(i)    Underestimation by models of OH concentrations measured in remote, biogenic environments: the global model predicts ROOOH peak concentrations in remote environments that are possibly high enough to explain, at least partially, the observed disagreement between model and measurements (Whalley et al., 2011;Lelieveld et al., 2008;Hofzumahaus et al., 2009;Tan et al., 2017).

(ii)   Variability of interferences observed in field campaigns: The box model calculations have shown that the concentration of ROOOH species varies with NO, VOC concentration and $J(O^1D)$ in the same way as the amplitude of the interference such as observed by the group of W. Brune (Feiner et al., 2016).

(iii)  Interference observed from $O_3$ + alkenes: the tentative explanation of alkene ozonolysis being the source of internally formed OH radicals through decomposition of the stabilized Criegee intermediate (Novelli et al., 2017) is possibly also, at least partially, due to ROOOH formed in a secondary reaction from $RO_2$ and OH, both generated during the ozonolysis (Johnson and Marston, 2008) of the very high VOC and $O_3$ concentrations in laboratory experiments (Novelli et al., 2014b;Rickly and Stevens, 2018;Fuchs et al., 2016). Indeed, it is observed in these experiments that the interference scales with the $O_3$+alkene turnover rate, i.e. the time that ROOOH can accumulate.

(iv)   Interferences observed in SAPHIR chamber: Fuchs *et al*. have carried out experiments under low NO conditions by comparing OH concentrations measured by FAGE and DOAS(Fuchs et al., 2012). Most of the time the agreement between both techniques was excellent, but on a few days towards the end of the campaign higher OH concentrations were measured by FAGE compared to DOAS. The NO concentrations on these days were lower, making the formation of ROOOH more likely, than on days with excellent agreement between FAGE and DOAS (Table 2 in Fuchs *et al*.(Fuchs et al., 2012)).

The results presented in this work thus propose a plausible solution to answer many open questions, it is however not very likely that it can explain an increase in the interference at night, such as observed by (Novelli et al., 2014a). Of course, the uncertainties are currently high on both, the observed FAGE interference per ROOOH molecule as well as the maximum ROOOH concentration that can accumulate in real environments. The first point could be improved through well-designed chamber studies under very low NO concentrations: such experiments have already been carried out (Nguyen et al., 2014)

and a detailed analysis of the data might support the conclusions from this work. The second point is more difficult to ameliorate because the steady state ROOOH concentration directly scales with its removal rate, and currently nothing is known about the fate of ROOOH. Perhaps the table can be turned by using the evolution of the observed interferences to learn about the fate of ROOOH?

Nonetheless, even with current uncertainties the implications on our understanding of daylight atmospheric oxidation chemistry are significant. We provide a plausible mechanism for how and why high OH levels in some environments are bolstered by a false signal, in a sense validating our current generation of models and reducing the need for speculative chemistry to explain the difference in simulated and observed OH of earlier field campaigns in pristine environments. With further observations and model development, the outcome will be to improve our ability to predict the OH budget in pristine

environments and the impacts of changes on the global chemistry-climate system.

**Author contribution**

SB and CS developed the FAGE instrument, CF and CS designed the experiments and MA carried them out., VF, SAN and ATA developed the model and performed the simulations. CF prepared the manuscript with contributions from all co-authors.

**Acknowledgements**

This project was supported by the French ANR agency under contract No. ANR-11-LabX-0005-01 CaPPA (Chemical and Physical Properties of the Atmosphere), the Région Hauts-de-France, the Ministère de l'Enseignement Supérieur et de la Recherche (CPER Climibio) and the European Fund for Regional Economic Development. ATA and SAN thank NERC-NCAS and the Walters-Kundert Trust under whose auspices this work was enabled. VF thanks the European Research

Council for funding through the Atmospheric Chemistry-Climate Interactions (ACCI) project, project number 267760. UM-UKCA runs in this work used the ARCHER UK National Supercomputing Service (http://www.archer.ac.uk). The authors thank P. Wennberg for very helpful discussions.

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
