# Peer review of "ROOOH: A missing piece of the puzzle for OH measurements in low NO environments?"

_Atmospheric Chemistry and Physics, 2018_

## Short Comment (SC1) · 26 Jun 2018

(A formatted version of this comment is attached as a supplement PDF)

———————————————————————

The results from this study are very interesting and could have wide reaching implications, not only for the understanding of atmospheric oxidation mechanisms, but also for the OH measurement community. The authors state that the possible interference from ROOOH decomposition in the FAGE apparatus could account for high OH concentration measurements around the globe from multiple groups. I am keen to understand more about the experiments and hence have some questions.

* Could the authors clarify their idea for the mechanism (chemical and/or physical) of

the decomposition of the ROOOH in the FAGE inlet?

* No two FAGE instruments are alike. Instruments where interference signals have been categorized have a range of different inlet lengths and inlet pinhole constructions (e.g. Faloona et al. (2004), Martinez et al. (2010) and Rickly and Stevens (2018)). Could the authors comment on the possible effects of FAGE instrument design on this interference?

* Were there any experiments conducted with different inlet pinhole diameters and inlet lengths to try and elucidate the effect on the possible ROOOH decomposition?

* Could the authors comment of the losses of ROOOH in the system? Are there expected losses on surfaces (e.g. the FAGE inlet pinhole)? Also, Müller et al. (2016), hypothesised a loss pathway for the ROOOH species via the reaction with water dimer. Will this be important under the experimental conditions presented here?

* The manuscript mentions the importance of OH scavenger experiments to determine whether there is a production of OH in the FAGE inlet (Novelli et al., 2014;Rickly and Stevens, 2018). Were similar experiments conducted here?
* * *
The flow tube experiments were conducted at high relative humidity (12000 ppmv) and with low flow rates to promote the formation of ROOOH over subsequent photolysis laser shots. I have a few questions about the chemistry and apparatus used. I think it would be useful information for the SOM for those in the FAGE community that might be interested:

* It has been hypothesized that RO2 radicals generated in the OH initiated oxidation of unsaturated hydrocarbons could form complexes with H2O molecules (Clark et al., 2010;Clark et al., 2008;Khan et al., 2015). Could the authors comment on the use of high [H2O] concentrations during these experiments and the possible effects this might have on the, already complicated, isoprene + OH oxidation mechanism?

* Was the effect of varying the initial conditions of the experiment investigated (e.g. $[H2O]0$, $[O3]0$ and $[OH]0$)? Was an alternative OH source used (without H2O)? Did the authors try other unsaturated hydrocarbons/terpenes such as pinene, as in Rickly and Stevens (2018)?

* Were more than two OH probe laser powers used in the determination of the absence of laser induced photolysis?

* The flow experiments were conducted in a regime where the photolysis beam did not fill the entire diameter of the flow tube. Could the authors comment on the possible impact of diffusion in and out of the photolysis region during the 20 s residence time?

* It would be useful to see an OH decay trace with pre-photolysis signal to judge the increase in the background level signal. Is the observed rise in the ILIF plateau above the S/N (and limit of detection) of the background signal, for example? Does the ILIF plateau value extend to the subsequent OH probe pulse? Was a run without an organic completed, to show the ILIF plateau base case in the absence of RO2 (and therefore ROOOH)?
* * *
Finally, I have a couple of comments about the conclusions:

* The contribution of ROOOH to [OH] measured with FAGE instruments depends highly on the production rate and loss processes for these molecules – both of which are highly uncertain at this stage.

* The interferences measured in the lab based ozonolysis experiments conducted by Novelli et al. (2014) and Rickly and Stevens (2018), have been shown to be removed upon addition of a reaction partner for Criegee intermediates (sulphur dioxide (Novelli et al., 2017) and acetic acid (Rickly and Stevens, 2018)). This suggests that stabilized Criegee intermediates decomposing in the FAGE inlet may be responsible in these cases.

\* The experiments conducted by Fuchs et al. (2012) did show a 30 – 40 % higher measurement of OH in a few cases, involving methyl vinyl ketone and toluene, which is indeed interesting. However, these runs were carried out under comparable NO conditions to other runs, for example containing isoprene, earlier in the campaign.
* * *
References:

Clark, J., English, A. M., Hansen, J. C., and Francisco, J. S.: Computational Study on the Existence of Organic Peroxy Radical-Water Complexes (RO2ÅůH2O), J. Phys. Chem. A, 112, 1587-1595, 10.1021/jp077266d, 2008.

Clark, J., Call, S. T., Austin, D. E., and Hansen, J. C.: Computational Study of Isoprene Hydroxyalkyl Peroxy Radical−Water Complexes (C5H8(OH)O2−H2O), J. Phys. Chem. A, 114, 6534-6541, 10.1021/jp102655g, 2010.

Faloona, I. C., Tan, D., Lesher, R. L., Hazen, N. L., Frame, C. L., Simpas, J. B., Harder, H., Martinez, M., di Carlo, P., Ren, X., and Brune, W. H.: A Laser-Induced Fluorescence Instrument for Detecting Troposhperic OH and HO2: Characteristics and Calibration, J. Atmos. Chem., 47, 139-167, 2004.

Fuchs, H., Dorn, H.-P., Bachner, M., Bohn, B., Brauers, T., Gomm, S., Hofzumahaus, A., Holland, F., Nehr, S., Rohrer, F., Tillmann, R., and Wahner, A.: Comparison of OH concentration measurements by DOAS and LIF during SAPHIR chamber experiments at high OH reactivity and low NO concentration, Atmos Meas Tech, 5, 1611-1626, 2012.

Khan, M. A. H., Cooke, M. C., Utembe, S. R., Archibald, A. T., Derwent, R. G., Jenkin, M. E., Morris, W. C., South, N., Hansen, J. C., Francisco, J. S., Percival, C. J., and Shallcross, D. E.: Global analysis of peroxy radicals and peroxy radical-water complexation using the STOCHEM-CRI global chemistry and transport model, Atmos. Environ., 106, 278-287, https://doi.org/10.1016/j.atmosenv.2015.02.020, 2015.

Martinez, M., Harder, H., Kubistin, D., Rudolf, M., Bozem, H., Eerdekens, G., Fischer, H., Klupfel, T., Gurk, C., Koenigstedt, R., Parchatka, U., Schiller, C. L., Stickler, A., Williams, J., and Lelieveld, J.: Hydroxyl radicals in the tropical troposphere over the Suriname rainforest: airborne measurements, Atmos. Chem. Phys., 10, 3759-3773, 10.5194/acp-10-3759-2010, 2010.

Müller, J.-F., Liu, Z., Nguyen, V. S., Stavrakou, T., Harvey, J. N., and Peeters, J.: The reaction of methyl peroxy and hydroxyl radicals as a major source of atmospheric methanol, Nat. Commun., 7, 13213, 10.1038/ncomms13213 https://www.nature.com/articles/ncomms13213#supplementary-information, 2016.

Novelli, A., Hens, K., Tatum Ernest, C., Kubistin, D., Regelin, E., Elste, T., Plass-Dülmer, C., Martinez, M., Lelieveld, J., and Harder, H.: Characterisation of an inlet pre-injector laser-induced fluorescence instrument for the measurement of atmospheric hydroxyl radicals, Atmos. Meas. Tech., 7, 3413-3430, 10.5194/amt-7-3413-2014, 2014.

Novelli, A., Hens, K., Tatum Ernest, C., Martinez, M., Nölscher, A. C., Sinha, V., Paasonen, P., Petäjä, T., Sipilä, M., Elste, T., Plass-Dülmer, C., Phillips, G. J., Kubistin, D., Williams, J., Vereecken, L., Lelieveld, J., and Harder, H.: Estimating the atmospheric concentration of Criegee intermediates and their possible interference in a FAGE-LIF instrument, Atmos. Chem. Phys., 17, 7807-7826, 10.5194/acp-17-7807-2017, 2017.

Rickly, P., and Stevens, P. S.: Measurements of a potential interference with laser-induced fluorescence measurements of ambient OH from the ozonolysis of biogenic alkenes, Atmos. Meas. Tech., 11, 1-16, 10.5194/amt-11-1-2018, 2018.

Please also note the supplement to this comment: https://www.atmos-chem-phys-discuss.net/acp-2018-441/acp-2018-441-SC1-supplement.pdf

---

## Author Comment (AC1) · 6 Jul 2018

The results from this study are very interesting and could have wide reaching implications, not only for the understanding of atmospheric oxidation mechanisms, but also for the OH measurement community. The authors state that the possible interference from ROOOH decomposition in the FAGE apparatus could account for high OH concentration measurements around the globe from multiple groups. I am keen to understand more about the experiments and hence have some questions.

* Could the authors clarify their idea for the mechanism (chemical and/or physical) of the decomposition of the ROOOH in the FAGE inlet?

The decomposition of ROOOH in the FAGE is not chemical; we suspect it to be a physical process probably due to decomposition caused by the fast variation of temperature within the shock wave during gas expansion, occurring when the mixture enters into the FAGE from high to low pressure. The ROOOH is stabilized by only around 120 kJ/mol with respect to $RO_2$ + OH, the decomposition to RO + $HO_2$ is slightly less endothermic (115 kJ/mol, see Assaf et al., IJCK DOI: 10.1002/kin.21191 (2018)), so one could also expect decomposition into $HO_2$. However, this cannot be tested in our set up, and well-designed experiments (for example in simulation chambers) will be necessary to test the mechanism for more details.

* No two FAGE instruments are alike. Instruments where interference signals have been categorized have a range of different inlet lengths and inlet pinhole constructions (e.g. Faloona et al. (2004), Martinez et al. (2010) and Rickly and Stevens (2018)). Could the authors comment on the possible effects of FAGE instrument design on this interference?

Yes we agree, the effect will probably not be the same for all FAGE systems. It is similar to the $RO_2$ interference on $HO_2$ measurements: each FAGE system has to be tested. We think the best test would be that the concerned groups re-analyze their data from earlier field campaigns in remote environments that have shown unexpected high OH concentrations: the reaction of $RO_2$ + OH could be integrated into their models and they could check if the disagreement between modeled and measured OH concentration correlates with the modeled ROOOH concentration. A "sensitivity" for each FAGE could be estimated from these tests, even though it is clear that this can be only relative, as currently nothing is known about the removal rate of ROOOH, and thus the resulting steady-state concentration.

* Were there any experiments conducted with different inlet pinhole diameters and inlet lengths to try and elucidate the effect on the possible ROOOH decomposition?

No measurements were conducted using different pinholes. As explained above, it is expected that this interference is instrument / pinhole / pump-speed / etc., specific and needs to be tested for each configuration used in the field. We think it is now clear (or even was already clear before) that all FAGE measurements in low NO environments must be carried out with the scavenger method. Our finding can give a possible explanation for the earlier unexplained measurements, and perhaps an advantage can be taken from this interference to use it for investigating the chemistry of ROOOH, just as the $RO_2$ problem in $HO_2$ detection was turned into a kind of advantage for quantifying $RO_2$ (see for example: Whalley LK, Stone D, Dunmore R, et al. Understanding in situ ozone production in the summertime through radical observations and modelling studies during the Clean air for London project (ClearfLo). *Atmos. Chem. Phys.* 2018;18:2547-2571).

* Could the authors comment of the losses of ROOOH in the system? Are there expected losses on surfaces (e.g. the FAGE inlet pinhole)? Also, Müller et al. (2016), hypothesised a loss pathway for the ROOOH species via the reaction with water dimer. Will this be important under the experimental conditions presented here?

No information on wall losses or reaction with the water dimer can be given due to the complexity of the production of the trioxide in our system. All we could do was to estimate by modeling what ROOOH concentration had probably been reached the FAGE. As we discuss in the manuscript, we think the model gives an upper limit, and in reality we have less ROOOH entering the FAGE due to wall losses and also to a decreased formation due to diffusion within the cell. But this is very difficult (if not impossible) to quantify. Also, our photolysis laser (quadrupled YAG) does not have a very homogeneous profile, i.e. there are some hot-spots. And as ROOOH is the product of a radical-radical reaction, this can make some difference in the final ROOOH concentration. Anyway, the goal of this paper is not a precise quantification of the interference, but rather to propose a new idea to solve some old questions.

* The manuscript mentions the importance of OH scavenger experiments to determine whether there is a production of OH in the FAGE inlet (Novelli et al., 2014;Rickly and Stevens, 2018). Were similar experiments conducted here?

We do add scavenger all the time: the isoprene, butane or $CH_4$ are all added in such high concentrations that all initial OH has reacted after a few 100 ms and we measure the remaining signal.

The flow tube experiments were conducted at high relative humidity (12000 ppmv) and with low flow rates to promote the formation of ROOOH over subsequent photolysis laser shots. I have a few questions about the chemistry and apparatus used. I think it would be useful information for the SOM for those in the FAGE community that might be interested:

* It has been hypothesized that $RO_2$ radicals generated in the OH initiated oxidation of unsaturated hydrocarbons could form complexes with $H_2O$ molecules (Clark et al., 2010;Clark et al., 2008;Khan et al., 2015). Could the authors comment on the use of high [$H_2O$] concentrations during these experiments and the possible effects this might have on the, already complicated, isoprene + OH oxidation mechanism?

With 12,000 ppm of $H_2O$ we have a relative humidity of around 50% at 20°C: this is the highest humidity that we can currently reach with our set-up. The $H_2O$ concentrations and relative humidity during the remote rainforest campaigns were certainly much higher. We have not carried out experiments under different $H_2O$ concentrations, because we need as much $H_2O$ as we can to make as many OH radicals as we can. Again, we think well-designed simulation chamber studies might be able to shine some light onto this question.

* Was the effect of varying the initial conditions of the experiment investigated (e.g. [$H_2O$]$_0$, [$O_3$]$_0$ and [OH]$_0$)? Was an alternative OH source used (without $H_2O$)? Did the authors try other unsaturated hydrocarbons/terpenes such as pinene, as in Rickly and Stevens (2018)?

As mentioned, we did not change $H_2O$ because the way the experiments are carried out requires very high OH concentration in order to generate high $RO_2$ concentration in order to allow competition between VOC and $RO_2$ after a few laser shots. Therefore, we have added

the highest $H_2O$ and $O_3$ possible. We have changed the VOC concentration in order to have 2 different conditions:

i)        High VOC concentration: no competition happens between the reaction of OH with the VOC and $RO_2$: in this case nearly all OH is consumed by reaction with VOC and high $RO_2$ concentrations are still formed, including the product of self-and cross reactions

ii)       Low VOC concentration: to have competition between the reaction of OH and the VOC leading to formation of trioxides

No other source of OH radicals was used. We have tested butane and methane because Assaf et al (2018) have measured the $HO_2$ yields for the reaction of these compounds with OH and have shown that the $HO_2$ yield in the case of methane is high, i.e. the expected ROOOH yield is low, while for butane the inverse is the case. Isoprene has been tested, because it is so important, even though nothing is known about the reaction of OH with the corresponding peroxy radical. Other VOCs are planned to be investigated in the future.

* Were more than two OH probe laser powers used in the determination of the absence of laser induced photolysis?

No, only 2 probe laser powers were tested: the highest possible, and a very low one, just high enough to get an exploitable S/N ratio. No energy in-between the two has been tested. To further test for a possible signal due to photolysis, we have also tested two different repetition rates of the excitation laser (see Figure S4)

* The flow experiments were conducted in a regime where the photolysis beam did not fill the entire diameter of the flow tube. Could the authors comment on the possible impact of diffusion in and out of the photolysis region during the 20 s residence time?

As explained above, diffusion has not been taken into account in the model, but would probably lead to a lower ROOOH concentration: considering diffusion leads to a decrease in average $RO_2$ concentration, and therefore the next pulse of OH radicals will see less $RO_2$ and more VOC (because fresh VOC from outside the photolysis volume diffuses into the photolysis volume). As mentioned, the given ROOOH concentration is an upper limit. On the other hand knowing the exact ROOOH concentration in the photolysis reactor is not very interesting or important to check for the possible interference in field campaigns, as the modeled ROOOH concentration directly depends on the removal rate of ROOOH in the atmosphere. And nothing is known about this number, i.e. an uncertainty range of a factor of 10 is certainly possible. The conclusion from the current work can only be that ROOOH is a very good candidate to explain the high OH concentrations observed in remote environments, but that's it, we cannot extract any more detailed numbers from our experiment.

* It would be useful to see an OH decay trace with pre-photolysis signal to judge the increase in the background level signal. Is the observed rise in the ILIF plateau above the S/N (and limit of detection) of the background signal, for example? Does the ILIF plateau value extend to the subsequent OH probe pulse? Was a run without an organic completed, to show the ILIF plateau base case in the absence of $RO_2$ (and therefore ROOOH)?

The pre-photolysis signal is the final signal of the preceding laser pulse, i.e. the value at 400 ms of pulse 1 corresponds to the signal 100 ms before pulse 2. The change in background is not visible by the eye (see figure 1 of the manuscript). Remember, only $\approx 4\times10^{-4}$ of ROOOH decomposed to OH radicals in our FAGE, i.e. the increase with respect to the initial OH signal is very small. However, the increase is statistically significant.

Carrying out a run without VOC will not be helpful, as the decay in zero air is too slow ($\approx 5$ s$^{-1}$) to allow for a precise determination of the plateau, i.e. the LIF signal does not decay fast enough between two laser pulses. However, experiments with high VOC concentrations (see figure S5 and S7) clearly show that, when there is no competition for OH radicals between VOC and $RO_2$ anymore, the background does not increase, even though the concentrations of all other products from cross- and self-reactions remain more or less the same. This test is much more useful to show that in the absence of ROOOH the plateau does not increase.
* * *
Finally, I have a couple of comments about the conclusions:

* The contribution of ROOOH to [OH] measured with FAGE instruments depends highly on the production rate and loss processes for these molecules – both of which are highly uncertain at this stage.

We fully agree: re-analyzing data from earlier field campaigns should be very interesting. As we show in the supplementary data, figure S9, the ROOOH steady state concentration scales linearly with the removal rate used in the model.

* The interferences measured in the lab based ozonolysis experiments conducted by Novelli et al. (2014) and Rickly and Stevens (2018), have been shown to be removed upon addition of a reaction partner for Criegee intermediates (sulphur dioxide (Novelli et al., 2017) and acetic acid (Rickly and Stevens, 2018)). This suggests that stabilized Criegee intermediates decomposing in the FAGE inlet may be responsible in these cases.

Maybe the addition of reaction partners for Criegee intermediates suppresses also the decomposition of Criegee within the reactor, and thus formation of OH and $RO_2$, and thus formation of ROOOH. But again, re-analysis of these data under this new aspect should be very interesting.

* The experiments conducted by Fuchs et al. (2012) did show a 30 – 40 % higher measurement of OH in a few cases, involving methyl vinyl ketone and toluene, which is indeed interesting. However, these runs were carried out under comparable NO conditions to other runs, for example containing isoprene, earlier in the campaign.

A close inspection of Table 2 in the Fuchs paper shows that on days with a slight disagreement between FAGE and DOAS the NO concentration has a tendency to be lower than on the other days. And as our model has shown, there is a very strong increase in ROOOH concentration at NO concentration below around 100 ppt, i.e. a difference of less than a factor of 2 in NO concentration might make a big difference in the interference. But again, this is only a hint and an idea; it should be very interesting to re-analyze these data sets under the aspect of including the reaction of $RO_2$ + OH into the models.

References:

Clark, J., English, A. M., Hansen, J. C., and Francisco, J. S.: Computational Study on the Existence of Organic Peroxy Radical-Water Complexes ($RO_2*H_2O$), J. Phys. Chem. A, 112, 1587-1595, 10.1021/jp077266d, 2008.

Clark, J., Call, S. T., Austin, D. E., and Hansen, J. C.: Computational Study of Isoprene Hydroxyalkyl Peroxy Radical−Water Complexes ($C_5H_8(OH)O_2$−$H_2O$), J. Phys. Chem. A, 114, 6534-6541, 10.1021/jp102655g, 2010.

Faloona, I. C., Tan, D., Lesher, R. L., Hazen, N. L., Frame, C. L., Simpas, J. B., Harder, H., Martinez, M., di Carlo, P., Ren, X., and Brune, W. H.: A Laser-Induced Fluorescence Instrument for Detecting Troposhperic OH and HO2: Characteristics and Calibration, J. Atmos. Chem., 47, 139-167, 2004.

Fuchs, H., Dorn, H.-P., Bachner, M., Bohn, B., Brauers, T., Gomm, S., Hofzumahaus, A., Holland, F., Nehr, S., Rohrer, F., Tillmann, R., and Wahner, A.: Comparison of OH concentration measurements by DOAS and LIF during SAPHIR chamber experiments at high OH reactivity and low NO concentration, Atmos Meas Tech, 5, 1611-1626, 2012.

Khan, M. A. H., Cooke, M. C., Utembe, S. R., Archibald, A. T., Derwent, R. G., Jenkin, M. E., Morris, W. C., South, N., Hansen, J. C., Francisco, J. S., Percival, C. J., and Shallcross, D. E.: Global analysis of peroxy radicals and peroxy radical-water complexation using the STOCHEM-CRI global chemistry and transport model, Atmos. Environ., 106, 278-287, https://doi.org/10.1016/j.atmosenv.2015.02.020, 2015.

Martinez, M., Harder, H., Kubistin, D., Rudolf, M., Bozem, H., Eerdekens, G., Fischer, H., Klupfel, T., Gurk, C., Koenigstedt, R., Parchatka, U., Schiller, C. L., Stickler, A., Williams, J., and Lelieveld, J.: Hydroxyl radicals in the tropical troposphere over the Suriname rainforest: airborne measurements, Atmos. Chem. Phys., 10, 3759-3773, 10.5194/acp-10-3759-2010, 2010.

Müller, J.-F., Liu, Z., Nguyen, V. S., Stavrakou, T., Harvey, J. N., and Peeters, J.: The reaction of methyl peroxy and hydroxyl radicals as a major source of atmospheric methanol, Nat. Commun., 7, 13213, 10.1038/ncomms13213 https://www.nature.com/articles/ncomms 13213#supplementary-information, 2016.

Novelli, A., Hens, K., Tatum Ernest, C., Kubistin, D., Regelin, E., Elste, T., Plass- Dülmer, C., Martinez, M., Lelieveld, J., and Harder, H.: Characterisation of an inlet preinjector laser-induced fluorescence instrument for the measurement of atmospheric hydroxyl radicals, Atmos. Meas. Tech., 7, 3413-3430, 10.5194/amt-7-3413-2014, 2014.

Novelli, A., Hens, K., Tatum Ernest, C., Martinez, M., Nölscher, A. C., Sinha, V., Paasonen, P., Petäjä, T., Sipilä, M., Elste, T., Plass-Dülmer, C., Phillips, G. J., Kubistin, D., Williams, J., Vereecken, L., Lelieveld, J., and Harder, H.: Estimating the atmospheric concentration of Criegee intermediates and their possible interference in a FAGE-LIF instrument, Atmos. Chem. Phys., 17, 7807-7826, 10.5194/acp-17-7807-2017, 2017.

Rickly, P., and Stevens, P. S.: Measurements of a potential interference with laserinduced fluorescence measurements of ambient OH from the ozonolysis of biogenic alkenes, Atmos. Meas. Tech., 11, 1-16, 10.5194/amt-11-1-2018, 2018.

---

## Referee Comment (RC1) · H. Harder (Referee) · 16 Jul 2018

Review acp-2018-441 Fittschen et al., 2018

ROOOH: the Missing Piece of the Puzzle for OH measurements in low NO Environments

The authors study an interference of the OH signal measured by an instrument based on the FAGE technique in a lab experiment where they photolysis a mixture of Ozone, water and isoprene or other VOCs with a varying number of laser pulses. After decay of the OH produced a non-zero signal remains, which increases with the number of photolysis laser pulses. The authors attribute this signal to trioxide (ROOOH) formed from recombination of RO2 and OH and claim this to be the interference observed by other LIF-FAGE instruments. They further analyze the potential abundance of ROOOH in the atmosphere by using the UM-UKCA global model.

The paper is concisely written. The idea of the potential role of ROOOH in contributing to the observed FAGE-LIF background OH as well as in the atmosphere is interesting and worth publication.

However, previous work is not adequately taken into consideration and contradicts some of the conclusions. Also, the authors base their conclusions on a less than 10% signal increase observed in their experiment and do not explain the remaining 90% of their signal. What is the cause of 90% of the signal ?

I recommend publication only after major revisions resolving this and the following issues.

*P2 L19, 23, 28….: "real" OH*

The OH at the time of fluorescence is real. Suggestion : Consider 'atmospheric' vs 'internally formed' or 'background' instead .

P2 L20 ff *Even though in practice this method is highly uncertain, given the generally low OH concentrations (and the resulting low S/N ratio) and the high temporal variability of OH radical concentration, the high OH concentrations observed in the different field campaigns seems to arise from "real" OH and not from the photolysis of other species.*

I would rather say the concentration of the background signal matters, not the signal of the OH concentration. The background signal found during HUMPPA 2010, Novelli et. al 2014, has a sufficiently high S/N, see. Fig. 14 in their paper demonstrates that the background signal observed does not show a square dependence on laser power.

*P2 L27: …the generation of OH radicals during the expansion into the FAGE cell…*

Please give citation and specify how do you define 'during the expansion' ? i.e. Do you refer to the cluster formation, shock front, evaporation phase ?

P2 L 33 : *This technique was used for the first time in 2012 in a forest in California (Mao et al., 2012)*

While all the credit of the using an OH scavenger mechanism should go to the group of Brune, Hens et. al 2014, reported measurements from a campaign in 2010. Novelli et al 2014 reported from campaigns conducted in 2010 (Finland & Spain), 2011(Germany), 2012 (Germany).

P3 L1: *fluorescence signal following scavenging of all ambient OH radicals, corresponding to up to 50% of the total OH concentration.*

Stating a relative contribution of the interference to a molecule that drops during nighttime close to zero is not meaningful. During nighttime, the relative contribution of the background signal is well above 90%. While during daytime it has been reported to be anything between 10-90%. I recommend to give equivalent OH mixing ratios/concentrations.

P3 L13ff: *Fuchs et al. (Fuchs et al., 2016) could not confirm this source: well below the detection limit of the FAGE.*

Not all LIF FAGE groups reported a significant interference, like the FAGE of the Jülich group, which does not observe a strong background signal in ambient air measurements in the first place. Extrapolation from instruments which do not see an interference in ambient air to those which do might not be valid.

P3 L21ff: *Following several years of interference studies in various environments, recent work from  W. Brune's  group (Feiner et al., 2016) concluded that the interference observed in their FAGE system; a) ;b);c); & d)*

P3 L28 *In this work we present convincing experimental and modelling evidence that this sought-after species is the product of the reaction between RO2.*

This might be true for the PennState instrument, though not for the Background of the Mainz instrument. a)-d) refer to the point given in the paper.

   a) *was due to a rather long-lived species because the interference  persists  into  the  evening,*
      Persistence into the night could be due to the persistence not just of the interfering species but of its precursors. Like O3 and terpenes, which are also temperature controlled and their emissions extend into the night.
   b)
   c) *it strongly increased with increasing O(1D), hence it must somehow be linked to photochemistry*
      The strong correlation with O1D is not the case, see Novelli et al. 2014, 2017; Mallik et al. 2018 the correlation is stronger with other parameters, like temperature, inverse of the square of the water vapor concentration, O3, concentration of some of the terpenes. Novelli et al. 2014, 2017 demonstrates the increase of the OH background signal during nighttime, when OH and JO1D are very low.
   d) *the species responsible for this interference  was  linked  to  a  low  NOx  oxidation  pathway, because  the  extent  of  the  interference  steeply  decreased  with  increasing NO concentration.*

There is no indication of an NO dependence on the abundance or production rate of the background OH in Novelli et al. 2017 or Mallik et al. 2018.

P5 L17 *This can be interpreted as interference…*

What about other possible second order isoprene oxidation products ?

P6 Fig2

While the increase of the signal with the number of pulses is consistent with this hypothesis, I find it disturbing that the signal remaining with just one pulse is already above 90% of the interference signal observed (0.051 according to Fig. 2) and is actually larger by an order of magnitude than the increase measured when the number of pulses is increased to 40 (which produces a signal of 0.056 according to Fig. 2). This makes it seem likely that there is an underlying interference not due to anything produced by the photolysing laser pulse, but present in the gas mixture even without photolysis.

No explanation is provided by the authors for the large signal after just one photolysis pulse. Also, no information is given about the signal observed when no photolysis pulse is applied, or when no VOC is added. Since ozone mixing ratios used of 600 ppb are high compared to ambient air, it should be stated whether there is a signal in humid air containing just VOC and ozone, or even just ozone, with and without a single photolysis pulse.

P10 L 9ff …*the observed disagreement between model and measurements*…

The model result does not reflect the abundance of the observations of the OH background signal reported by Novelli et al 2014 & 2017  as well as Mallik et al. 2018. The relative contributions observed is largest during summer in the boreal forest of Finland 2010, larger than on Cyprus 2014 or southern Germany 2012.

P10 L 13 ff : *Variability of interferences observed in field campaigns*

This might be valid for the PennState group, but the data reported by Novelli, etc do not support the dependence on JO1D and low NOx conditions.

---

## Referee Comment (RC2) · Anonymous Referee #2 · 8 Aug 2018

The paper is an interesting one, and suggests that ROOOH, present in the atmosphere from the reaction of RO2 with OH, can somehow generate OH within the inlets and fluorescence chambers of instruments that use laser-induced fluorescence to determine OH levels in the atmosphere. Results are shown for the Lille system, where a signal is seen from a mixture of isoprene and an OH precursor (O3/H2O/hv) when the RO2 + OH reaction is initiated within a flow-tube that is sampled by the FAGE instrument. The paper then discusses that ROOOH present in the atmosphere may act as a source of interference for OH measurements in order to explain some previous model/measurement discrepancies.

The results are interesting, and I agree that ROOOH decomposition within the inlet or cell of the instrument should certainly be considered as a possibility for generating

some artificial OH signal. Nowadays FAGE instruments by some groups are operated with a scavenger inlet, which would enable any OH from ROOOH to be allowed for, so this finding is more relevant to previous field measurements without a scavenger inlet where the signal may partly be from an interfering species.

The range of experiments performed certainly seems to show that there is OH signal at long times (which grows after exposure of the OH reactivity flow tube to multiple photolysis laser shots) which is only present for the longer chain RO2 (butane and also isoprene as the VOCs) where the ROOOH yield is expected to be larger. The RO2+OH reaction until recently was largely overlooked as important in the atmosphere, but work by the Lille group, and more recently from a US group, have measured the rate coefficient. For R=CH3 there is still a significant disagreement in the rate coefficient (the disagreement was initially worse, but a revised rate coefficient from Lille brought the values closer to each other). Lille have studied RO2+OH now for a number of R, and have measured the HO2 yield, which decreases for larger R (Assaf et al., IJCK 2018), providing evidence that the yield of ROOOH increases as R gets bigger. Recently the yield of CH3OH from CH3O2+OH was quantified experimentally to be small (there had been some theory on this reaction regarding the CH3OH yield).

Although there is clearly a signal in the Lille FAGE system at long times which grows further with the number of photolysis shots and which may derive from the ROOOH product of RO2+OH, it is surely a considerable stretch to say that this constitutes convincing evidence that the disagreement between model and measurements seen in previous field campaigns (under relevant conditions like low NOx forests where ROOOH is expected to have higher concentrations) is due to interference by the decomposition of ROOOH, and that ROOOH reflects the missing piece of the puzzle. There are several parameters which are unknown or poorly known, the most important of which is the concentration of ROOOH in the atmosphere, another is the fraction of ROOOH which may decompose within the fluorescence inlet/cell (which will be instrument dependent). In order for the atmospheric model in this paper to generate

something that would generate a relevant level of OH, a decomposition fraction is assumed (based on an estimate from the Lille lab. data) and the loss rate of ROOOH was varied over 3 orders of magnitude and a value chosen to give a level of OH similar to atmospheric levels.

Another major point is even if ROOOH in the atmosphere were to constitute an interference that needed consideration, different FAGE instruments behave differently when it comes to interference formed in the inlet / fluorescence cell. This depends on a variety of factors including the residence time inside the instrument, inlet length from sampling pinhole to fluorescence cell, cell geometry (size, distance from walls), fluorescence imaging volume (e.g. single pass or multi-pass as the case here) and others like the pumping rate. For the case of HO2 interference from RO2 for example, the residence time and the geometry of the sampling/cell system has been shown to control the level of interference shown. This needs to be stated in the paper.

Concerning the mechanism of the decomposition of ROOOH within the low-pressure FAGE cell, it is not clear what the mechanism is? It is colder (initially in the expansion) and the number density is low so what is the source of the energy? Is it a homogeneous or a heterogeneous process? For a gas phase process the energetics (Assaf et al., IJCK 2018) seem to suggest if decomposition occurs this would be via RO + HO2 rather than RO2 +OH? As the energetics are known can the ratio of RO2+OH to RO +HO2 be calculated? Is the ROOOH decomposing at all in the OH reactivity flow-tube (which is at atmospheric pressure – presumably any OH is then very quickly removed but could decomposition sustain a small steady-state level).

There will be mixing in of non-photolysed gas where 266 nm is not present – and so not all of the gas will have been illuminated by the same number of pulses? Also, would the 266 nm radiation generate any OH from hitting the metal pinhole at the end of the flow-tube?

The OH signal does not decay to zero at long times for the isoprene (and to a lesser

extent butane) and the long time signal (plateau signal) increases for the number of 266 nm laser shots – and this is the main evidence that OH (signal in the FAGE cell) builds up with time as more ROOOH is made in the OH reactivity flowtube for longer exposure as the isoprene gets used up. Monitoring the OH reactivity with the number of photolysis shots is a clever idea, and it clearly decreases as the isoprene gets used up. It would be nice to see the t<0 baseline level to compare with the long time plateau signal. The t<0 signal is not shown for any of the plots. For Figure S8 for methane, there is no build up of the long time signal with the number of pulses (evidence for no ROOOH being formed for R=CH3), but the signal is not zero? Presumably the t<0 signal should be zero? Why is there a "constant" underlying signal at long times, even for just 1 photolysis pulse there seems to be a long time signal, i.e. for Figure 2, for 1 photolysis pulse the plateau value is around 0.05 or so (right axis) and it only increases to 0.055 or so after 40 pulses?

There are some processes not considered in the model, namely RO2, isoprene (products) and ROOOH photolysis at 266 nm, which will generate products that may undergo secondary chemistry (e.g. second order isoprene oxidation products?) and build up something which makes OH in the FAGE cell?

If the photolysis is stopped after say 20 shots, could the loss of ROOOH from the flowtube be monitored in some way?

For the determination of the OH reactivity, was the same fitting window used for all of the decays, i.e. was it the same for the Pulse 1 decay as for the Pulse 40 decay? Or was a different start and end time used?

There is no doubt that the work is interesting, and the idea that ROOOH decomposing to OH and hence constituting a potential interference for OH instruments is worthwhile, and needs further exploration. However, it is unknown what the mechanism of decomposition might be, and it is a complex chemical system and the model used to explain the laboratory results has many assumptions/simplifications – and there may be other

species generated in the system which could release OH once inside the FAGE cell. There is likely to be a cascade of chemistry over the 40 photolysis shots generating many species (some of the products and intermediates formed will be different to the real atmosphere owing to the conditions and wavelength of light present).

The language in the paper needs moderating considerably regarding extrapolating the laboratory observations to the conclusion that the work presents convincing evidence that previous model-measurement uncertainties in low NOx environments containing isoprene is due to ROOOH (R from isoprene or similar) decomposition. Our knowledge of ROOOH abundance and removal processes is virtually non-existent, and the model has been optimised using adjustable parameters so as to give a level of OH in the instrument which becomes important compared with atmospheric levels. The wording over the implications of the work, and also the title of the paper, needs changing.

Other points:

Title, abstract and conclusions – these need to be toned down to report the observations made, that ROOOH needs to be considered as a potential source of OH inside instruments. Any implication that the puzzle is solved is going too far and is rather premature. The title cannot remain as is. The abstract and conclusion also need to make it clear that any OH from ROOOH would be highly dependent upon the instrument design, and make implicit any assumptions made in the model.

Words such as "convincing" are of course subjective – and should be toned down or removed completely.

The balance of the main paper and the supplementary information seems skewed. There is information in the SI which really ought to be in the main paper (there is no compelling reason this has to be done for space reasons). e.g. Figures S1 and S4, S6 (this seems central to show) and probably one panel for butane and methane.

Page 4, line 18, it should be 200 microseconds (not ms).

Figure 1. It is not clear how the t=0 maximum in OH signal varies with the number of shots? Does this change with pulse 1 to 40 in any systematic way? Also what does "data analysis is carried out with raw data" mean?

Figure 2. The red line shows the plateau OH signal from the fit, gradually increasing with the number of photolysis pulses. Given the very small magnitude of the plateau OH compared with the initial OH signal the accuracy of the fit at long time is very important. Could Figure 1 at long times be plotted on a very expanded vertical scale (perhaps from y=0 to y=0.15) and the fits be shown with it? The results are very dependent on how well things fit at long times. Related to this point I would like to see Figure 2 plotted when the plateau is averaged using the data at later times. As the OH reactivity becomes smaller with the number of photolysis pulses, it takes longer for the signal to reach the plateau, so does this slower decay promote the increasing baseline signal in any way? There are certainly some decays in the SI that do not seem to have reached the baseline before the averaging window to obtain the plateau signal begins.

Page 6, lines 5-15. Excitation laser energy and photolysis energy are both used here – be clearer about which pulse energy was changed. Clearly Fig S2 and S3 are for probe (excitation) laser energies, but unclear if photolysis (266 nm) energy was changed also?

The error in [ROOOH] from the model after 40 shots (figure S6 model) needs to be stated.

Why was 10(-4) s-1 chosen as the loss rate for ROOOH in the atmosphere? Presumably as this gave an OH concentration of around 1x10(6)? There is a very large uncertainty in [ROOOH], and so the statement on page 9, line 8 that [ROOOH] is predicted to be of the order of 50-200 pptv seems very optimistic in terms of the range of concentrations? For Figure 4, were there RO2 measurements made in the field to constrain the model, or was modelled RO2 used?

Discussion

Page 10, line 6.

It is stated that the product of the RO2+OH reaction leads to an OH interference. The signal may originate from that, but there may be other origins of the signal also, and so the wording needs to be more flexible. This is also where a statement about the different FAGE designs is needed.

Supplementary material.

I think much of section 1 on the FAGE and probe system needs to go in the main paper.

What is the residence time of the FAGE detection cell (inlet from the pinhole and fluorescence cell)? There is no discussion of this and it is an important point. Was the pumping rate of the cell changed to change the residence time?

Figure S2 – in the caption add a line saying what the black line is.

Line 187 of SI – "is consistent with" is better than "is indeed due to"

For all of the OH signal versus time plots the y axis ends at zero. Are there any negative points, or is the "baseline" some value a bit above zero?

Figure S6. The concentrations versus no of photolysis pulses. This seems central and should be in the main paper.

Line 220, reword "very basically"

---

## Author Comment (AC2) · 3 Sep 2018

ROOOH: the Missing Piece of the Puzzle for OH measurements in low NO Environments
The authors study an interference of the OH signal measured by an instrument based on the FAGE technique in a lab experiment where they photolysis a mixture of Ozone, water and isoprene or other VOCs with a varying number of laser pulses. After decay of the OH produced a non-zero signal remains, which increases with the number of photolysis laser pulses. The authors attribute this signal to trioxide (ROOOH) formed from recombination of $RO_2$ and OH and claim this to be the interference observed by other LIF-FAGE instruments. They further analyze the potential abundance of ROOOH in the atmosphere by using the UM-UKCA global model.

The paper is concisely written. The idea of the potential role of ROOOH in contributing to the observed FAGE-LIF background OH as well as in the atmosphere is interesting and worth publication. However, previous work is not adequately taken into consideration and contradicts some of the conclusions. Also, the authors base their conclusions on a less than 10% signal increase observed in their experiment and do not explain the remaining 90% of their signal. What is the cause of 90% of the signal?

The remaining 90% of the signal is due to stray light from the excitation laser as well as ambient light entering through the photolysis window and the nozzle. This has now been explained Page 4. The conclusion of this work is based on the fact that we observe an increase in this background signal only under conditions where $RO_2$ will partially react with OH, under any other condition the background is stable.

I recommend publication only after major revisions resolving this and the following issues.

*P2 L19, 23, 28....: "real" OH*

The OH at the time of fluorescence is real. Suggestion: Consider 'atmospheric' vs 'internally formed' or 'background' instead.

We have changed "real' into "atmospheric"

P2 L20 ff *Even though in practice this method is highly uncertain, given the generally low OH concentrations (and the resulting low S/N ratio) and the high temporal variability of OH radical concentration, the high OH concentrations observed in the different field campaigns seems to arise from "real" OH and not from the photolysis of other species.*

I would rather say the concentration of the background signal matters, not the signal of the OH concentration. The background signal found during HUMPPA 2010, Novelli et. al 2014, has a sufficiently high S/N, see. Fig. 14 in their paper demonstrates that the background signal observed does not show a square dependence on laser power.

Page 2 we have added:

*This was also confirmed by Novelli et al (Novelli et al., 2014a) who observed a strong background during HUMPPA2010 with good S/N ratio, allowing to unequivocally exclude photolysis being at the origin of the background signal.*

*P2 L27: ...the generation of OH radicals during the expansion into the FAGE cell...*
Please give citation and specify how do you define 'during the expansion' ? i.e. Do you refer to the cluster formation, shock front, evaporation phase ?

*Indeed, we do not know how exactly the ROOOH leads to formation of OH radicals in our FAGE system. We have shown that is it NOT due to photolysis, but for the rest, we don't know. It could be thermal decomposition within the shock front, however one would expect mostly (or even exclusively) a decomposition to $HO_2$ and RO, given that this decomposition path is around 10 kJ $mol^{-1}$ cheaper. According to Müller et al (sup data), the thermal decomposition of ROOOH is very slow and not a major fate of this species under atmospheric conditions. Also, they estimate the exclusive reaction products being $CH_3O + HO_2$ and $CH_3OH + O_2$. It is therefore rather unlikely that the ROOOH decomposes thermally within the shockwave, or even in the photolysis reactor, leading to a low steady-state concentration. Maybe the decomposition is heterogeneous, either on the nozzle or on the wall of the FAGE cell. This would mean of course even more that different FAGE instruments will show different sensitivity to ROOOH. We have added at the end of the experimental part:*

*No clear explanation can be given on the mechanism of this OH formation: a homogeneous decomposition within the shock wave of the expansion is unlikely, because the pathway leading to $CH_3O$ and $HO_2$ is thermodynamically more favoured (Assaf et al., 2018a). Therefor a heterogeneous decomposition on the walls of the FAGE cell or the entrance nozzle are more likely. The residence time of the gas mixture between entrance nozzle and detection beam can be calculated from the volume of the cell (0.25 l) and the gas flow (3 l $min^{-1}$ STP) to around 1 sec, leaving ample time for collisions with the reactor walls.*

P2 L 33 : *This technique was used for the first time in 2012 in a forest in California (Mao et al., 2012)* While all the credit of the using an OH scavenger mechanism should go to the group of Brune, Hens et. al 2014, reported measurements from a campaign in 2010. Novelli et al 2014 reported from campaigns conducted in 2010 (Finland & Spain), 2011(Germany), 2012 (Germany).

*In order to keep to credit of having "invented' this technique and still being precise, and given that your paper appeared 2 years after Mao et al., we have changed the sentence to the following:*

*The use of this technique was **reported** for the first time in 2012 showing results for a field campaign in a forest in California (Mao et al., 2012). It led to the identification of a large fluorescence signal following scavenging of all ambient OH radicals, corresponding to up to 50% of the total OH concentration.*

P3 L1: *fluorescence signal following scavenging of all ambient OH radicals, corresponding to up to 50% of the total OH concentration.*

Stating a relative contribution of the interference to a molecule that drops during nighttime close to zero is not meaningful. During nighttime, the relative contribution of the background signal is well above 90%. While during daytime it has been reported to be anything between 10-90%. I recommend to give equivalent OH mixing ratios/concentrations.

*We have now linked this information to the paper of Mao and not as a general statement on observed interferences.*

P3 L13ff: *Fuchs et al. (Fuchs et al., 2016) could not confirm this source: well below the detection limit of the FAGE.*

Not all LIF FAGE groups reported a significant interference, like the FAGE of the Jülich group, which

does not observe a strong background signal in ambient air measurements in the first place. Extrapolation from instruments which do not see an interference in ambient air to those which do might not be valid.

We do not claim that all FAGE instruments behave the same. This has now been emphasized on several occasions, and also in the abstract. However, there are hints that the Jülich FAGE also suffers from some interference in the OH measurements. First there is the still unexplained high OH measurements from the PRIDE campaign with the disagreement correlating with decreased NO concentration, in line with the present hypothesis. Second, they have recently (Tan et al. 2017) used for the first time a pre-injector prototype and have observed some unexplained OH, even though some technical issues made the measurements uncertain. However, taking from the 6 values in Table 2 only the 4 values at low NO, then the unexplained OH (divided by the total OH concentration to "normalize" to overall photochemical activity) increases with decreasing NO, also in line with the current hypothesis of ROOOH being source of OH in FAGE instruments.

[Figure]

Finally, as we already replied to Winiberg: "A close inspection of Table 2 in the Fuchs paper shows that on days with a slight disagreement between FAGE and DOAS the NO concentration has a tendency to be lower than on the other days. And as our model has shown, there is a very strong increase in ROOOH concentration at NO concentration below around 100 ppt, i.e. a difference of less than a factor of 2 in NO concentration might make a big difference in the interference. But again, this is only a hint and an idea; it should be very interesting to re-analyze these data sets under the aspect of including the reaction of $RO_2$ + OH into the models."

But: the reaction of $O_3$ + alkene does not lead to an interference in their FAGE that is high enough to explain these observations. But again, we do not say that the reaction between $O_3$ and alkene cannot be a source of interference in your FAGE.

P3 L21ff: *Following several years of interference studies in various environments, recent work from W. Brune's group (Feiner et al., 2016) concluded that the interference observed in their FAGE system; a) ;b);c); & d)*

P3 L28 *In this work we present convincing experimental and modelling evidence that this sought-after species is the product of the reaction between $RO_2$.*

This might be true for the PennState instrument, though not for the Background of the Mainz instrument. a)-d) refer to the point given in the paper.

Yes, point a-d refer only to the observations from the Brune group (line 22: in **their** FAGE system), we do not state anything about the background observed with your instrument. It is of course possible that there are different sources of interference for different instruments and for different conditions. As mentioned already, this has now been emphasized at different occasions.

a) *was due to a rather long-lived species because the interference persists into the evening,* Persistence into the night could be due to the persistence not just of the interfering species but of its precursors. Like $O_3$ and terpenes, which are also temperature controlled and their emissions extend into the night.

b)
c) *it strongly increased with increasing $O(^1D)$, hence it must somehow be linked to photochemistry* The strong correlation with $O^1D$ is not the case, see Novelli et al. 2014, 2017; Mallik et al. 2018 the correlation is stronger with other parameters, like temperature, inverse of the square of the water vapor concentration, $O_3$, concentration of some of the terpenes. Novelli et al. 2014, 2017 demonstrates the increase of the OH background signal during nighttime, when OH and $J(O^1D)$ are very low.

d) *the species responsible for this interference was linked to a low $NO_x$ oxidation pathway, because the extent of the interference steeply decreased with increasing NO concentration.* There is no indication of an NO dependence on the abundance or production rate of the background OH in Novelli et al. 2017 or Mallik et al. 2018.

Indeed, Novelli et al. 2017 or Malik et al 2018 did not show any dependence of the background signal on NO. However, comparison of the OH background signal measured during the HUMPPA-campaign (Figure 8 in Novelli et al. 2014) with the NO concentration data (Figure 9 of Hens et al. 2014) show, that during the last days of the campaign (August 7 and 8), when NO is extremely low, the ratio of atmospheric OH to background OH is much higher than in the beginning of the campaign, when NO increases above 100 ppt (unfortunately, OH data are not complete, especially during the "high" NO periods). For example, on August 2 and 5, NO reaches 200 ppt and the background makes only roughly 50% of the total signal, while on August 7 and 8 nearly 100% of the OH signal is due to background and NO is well below 100ppt on these days. But again, only a detailed re-analysis of the data taking into account the reaction of $RO_2$ + OH can show if part of the background signal might be correlated with the turnover of the reaction of $RO_2$ radicals with OH.

P5 L17 *This can be interpreted as interference…*

What about other possible second order isoprene oxidation products?

In order to distinguish between the products of the reaction of $RO_2$ + OH and other oxidation products, we have carried out experiments for isoprene and butane with increased VOC concentration, but the same OH and $O_3$ concentrations: under these conditions the mixture contains nearly equal concentrations of all other oxidation products, except for $RO_2$ + OH. Under these conditions, the background is stable (Figure S5 and S7, now in the main text). From this observation we conclude that

the rise in background is due to the product of $RO_2 + OH$ and not from any other second order isoprene oxidation product.

P6 Fig2

While the increase of the signal with the number of pulses is consistent with this hypothesis, I find it disturbing that the signal remaining with just one pulse is already above 90% of the interference signal observed (0.051 according to Fig. 2) and is actually larger by an order of magnitude than the increase measured when the number of pulses is increased to 40 (which produces a signal of 0.056 according to Fig. 2). This makes it seem likely that there is an underlying interference not due to anything produced by the photolysing laser pulse, but present in the gas mixture even without photolysis. No explanation is provided by the authors for the large signal after just one photolysis pulse. Also, no information is given about the signal observed when no photolysis pulse is applied, or when no VOC is added. Since ozone mixing ratios used of 600 ppb are high compared to ambient air, it should be stated whether there is a signal in humid air containing just VOC and ozone, or even just ozone, with and without a single photolysis pulse.

As explained above, the signal present already even at the first photolysis shot is due to laser stray light and ambient light entering through photolysis window and pinhole. As given already in our answer to Winiberg, a signal with only $O_3$ is not useful because the OH signal does decay much too slow to reliably measure the background. Again, the most convincing test is the experiment with high VOC, showing that the background does not increase under these conditions. We have now added in Figure 1 (now figure 2) the pre-photolysis signal as well as a zoom on the background signal for the different pulses.

P10 L 9ff …*the observed disagreement between model and measurements*…

The model result does not reflect the abundance of the observations of the OH background signal reported by Novelli et al 2014 & 2017 as well as Mallik et al. 2018. The relative contributions observed is largest during summer in the boreal forest of Finland 2010, larger than on Cyprus 2014 or southern Germany 2012.

We think that in saying: "**If** occurring also with a comparable intensity in other FAGE instruments, it **can** be high enough …." we are careful. To more emphasize this fact, we have added :

*The intensity or even the occurrence at all can depend on the design and working conditions of the FAGE set-up, which is different for different groups. However, if occurring also with a comparable intensity in other FAGE instruments, this interference might be high enough to explain numerous observations obtained with FAGE instruments from other research groups including: ……*

In any case, we do not state that the observations with your FAGE can be explained by ROOOH (even if they might).

P10 L 13 ff : *Variability of interferences observed in field campaigns*

This might be valid for the PennState group, but the data reported by Novelli, etc do not support the dependence on $J(O^1D)$ and low $NO_x$ conditions.

This is what we say: "…. such as observed by the group of W. Brune"

---

## Author Comment (AC3) · 3 Sep 2018

The paper is an interesting one, and suggests that ROOOH, present in the atmosphere from the reaction of $RO_2$ with OH, can somehow generate OH within the inlets and fluorescence chambers of instruments that use laser-induced fluorescence to determine OH levels in the atmosphere. Results are shown for the Lille system, where a signal is seen from a mixture of isoprene and an OH precursor ($O_3/H_2O/hv$) when the $RO_2$ + OH reaction is initiated within a flow-tube that is sampled by the FAGE instrument. The paper then discusses that ROOOH present in the atmosphere may act as a source of interference for OH measurements in order to explain some previous model/measurement discrepancies.

The results are interesting, and I agree that ROOOH decomposition within the inlet or cell of the instrument should certainly be considered as a possibility for generating some artificial OH signal. Nowadays FAGE instruments by some groups are operated with a scavenger inlet, which would enable any OH from ROOOH to be allowed for, so this finding is more relevant to previous field measurements without a scavenger inlet where the signal may partly be from an interfering species.

The range of experiments performed certainly seems to show that there is OH signal at long times (which grows after exposure of the OH reactivity flow tube to multiple photolysis laser shots) which is only present for the longer chain $RO_2$ (butane and also isoprene as the VOCs) where the ROOOH yield is expected to be larger. The $RO_2$ + OH reaction until recently was largely overlooked as important in the atmosphere, but work by the Lille group, and more recently from a US group, have measured the rate coefficient. For R=$CH_3$ there is still a significant disagreement in the rate coefficient (the disagreement was initially worse, but a revised rate coefficient from Lille brought the values closer to each other). Lille have studied $RO_2$+OH now for a number of R, and have measured the $HO_2$ yield, which decreases for larger R (Assaf et al., IJCK 2018), providing evidence that the yield of ROOOH increases as R gets bigger. Recently the yield of $CH_3OH$ from $CH_3O_2$+OH was quantified experimentally to be small (there had been some theory on this reaction regarding the $CH_3OH$ yield).

Although there is clearly a signal in the Lille FAGE system at long times which grows further with the number of photolysis shots and which may derive from the ROOOH product of $RO_2$+OH, it is surely a considerable stretch to say that this constitutes convincing evidence that the disagreement between model and measurements seen in previous field campaigns (under relevant conditions like low NOx forests where ROOOH is expected to have higher concentrations) is due to interference by the decomposition of ROOOH, and that ROOOH reflects the missing piece of the puzzle. There are several parameters which are unknown or poorly known, the most important of which is the concentration of ROOOH in the atmosphere, another is the fraction of ROOOH which may decompose within the fluorescence inlet/cell (which will be instrument dependent). In order for the atmospheric model in this paper to generate something that would generate a relevant level of OH, a decomposition fraction is assumed (based on an estimate from the Lille lab. data) and the loss rate of ROOOH was varied over 3 orders of magnitude and a value chosen to give a level of OH similar to atmospheric levels.

Another major point is even if ROOOH in the atmosphere were to constitute an interference that needed consideration, different FAGE instruments behave differently when it comes to interference formed in the inlet / fluorescence cell. This depends on a variety of factors including the residence time inside the instrument, inlet length from sampling pinhole to fluorescence cell, cell geometry (size, distance from walls), fluorescence imaging volume (e.g. single pass or multi-pass as the case here) and others like the pumping rate. For the case of $HO_2$ interference from $RO_2$ for example, the residence time and the geometry of the sampling/cell system has been shown to control the level of interference shown. This needs to be stated in the paper.

We completely agree that the effect might be very different in different FAGE systems. It was already mentioned in the beginning of the discussion that the results are only valid for the UL-FAGE (Page 10: If occurring also with a comparable intensity in other FAGE instruments, it……), but we have now emphasized this fact on several occasions (Page 4, line 9ff and Page 7, line 26ff) as well as in the abstract.

Concerning the mechanism of the decomposition of ROOOH within the low-pressure FAGE cell, it is not clear what the mechanism is? It is colder (initially in the expansion) and the number density is low so what is the source of the energy? Is it a homogeneous or a heterogeneous process? For a gas phase process the energetics (Assaf et al., IJCK 2018) seem to suggest if decomposition occurs this would be via $RO + HO_2$ rather than $RO_2 + OH$? As the energetics are known can the ratio of $RO_2 + OH$ to $RO + HO_2$ be calculated? Is the ROOOH decomposing at all in the OH reactivity flow-tube (which is at atmospheric pressure – presumably any OH is then very quickly removed but could decomposition sustain a small steady-state level).

Indeed, we do not know how exactly the ROOOH leads to formation of OH radicals in our FAGE system. We have shown that is it NOT due to photolysis, but for the rest, we don't know. It could be thermal decomposition within the shock front, however one would expect mostly (or even exclusively) a decomposition to $HO_2$ and RO, given that this decomposition path is around 10 kJ mol$^{-1}$ more energetically favourable. According to Müller et al (in their supplementary data), the thermal decomposition of ROOOH is very slow and not a major fate of this species under atmospheric conditions. Also, they estimate the exclusive reaction products being $CH_3O + HO_2$ and $CH_3OH + O_2$. It is therefore rather unlikely that the ROOOH decomposes thermally within the shockwave, or even in the photolysis reactor, leading to a low steady-state concentration. Maybe the decomposition is heterogeneous, either on the nozzle or on the wall of the FAGE cell. This would mean of course even more that different FAGE instruments would show different sensitivity to ROOOH. We have added at the end of the experimental part:

*No clear explanation can be given on the mechanism of this OH formation: a homogeneous decomposition within the shock wave of the expansion is unlikely, because the pathway leading to $CH_3O$ and $HO_2$ is thermodynamically more favoured (Assaf et al., 2018a). Therefor a heterogeneous decomposition on the walls of the FAGE cell or the entrance nozzle are more likely. The residence time of the gas mixture between entrance nozzle and detection beam can be calculated from the volume of the cell (0.25 l) and the gas flow (3 l min$^{-1}$ STP) to around 1 sec, leaving ample time for collisions with the reactor walls.*

There will be mixing in of non-photolysed gas where 266 nm is not present – and so not all of the gas will have been illuminated by the same number of pulses? Also, would the 266 nm radiation generate any OH from hitting the metal pinhole at the end of the flow-tube?

The diameter of the photolysis beam is 2.5 cm while the reactor has a diameter of 5 cm. We have not considered in our model that a part of the mixture will not be illuminated and that diffusion will dilute the mixture. This will add to the uncertainty of the estimated ROOOH concentration and lead to an overestimation, given that the key reaction is a radical-radical reaction and relies on the concentration of $RO_2$ generated in the previous laser pulse, which will have time between two pulses to diffuse into the non-illuminated volume.

Generation of OH from 266nm light hitting the pinhole would probably have no impact on the signal at long reaction times and only influence the signal shortly after the laser pulse.

The OH signal does not decay to zero at long times for the isoprene (and to a lesser extent butane) and the long time signal (plateau signal) increases for the number of 266 nm laser shots – and this is the main evidence that OH (signal in the FAGE cell) builds up with time as more ROOOH is made in the OH reactivity flowtube for longer exposure as the isoprene gets used up. Monitoring the OH reactivity with the number of photolysis shots is a clever idea, and it clearly decreases as the isoprene gets used up. It would be nice to see the t<0 baseline level to compare with the long time plateau signal. The t<0 signal is not shown for any of the plots. For Figure S8 for methane, there is no build up of the long time signal with the number of pulses (evidence for no ROOOH being formed for R=CH$_3$), but the signal is not zero? Presumably the t<0 signal should be zero? Why is there a "constant" underlying signal at long times, even for just 1 photolysis pulse there seems to be a long time signal, i.e. for Figure 2, for 1 photolysis pulse the plateau value is around 0.05 or so (right axis) and it only increases to 0.055 or so after 40 pulses?

The t<0 signal is not zero due to stray light from the excitation laser as well as some ambient laboratory light entering through the photolysis window and the nozzle. This has now been explained on Page 4. The conclusion of this work is based on the fact that we observe an increase in this background signal only under conditions where RO$_2$ will partially react with OH; under any other condition the background is stable. We have now zoomed in Figure 2 onto the background, and have also shown the t<0 signal (we measure 15 ms before the laser pulse), even though this signal is at 2 Hz repetition rate nearly equivalent to the data points at long reaction time: there are only 100 ms without data points between the end of one trace and the beginning of the next one.

There are some processes not considered in the model, namely RO$_2$, isoprene (products) and ROOOH photolysis at 266 nm, which will generate products that may undergo secondary chemistry (e.g. second order isoprene oxidation products?) and build up something which makes OH in the FAGE cell?

Except for the photolysis of ROOOH all other processes will also take place in the same way when we do the experiments with higher VOC concentrations. And one of the main evidences of our hypothesis is the fact that we do NOT observe an increase in the background when we work under conditions (high VOC concentration) where all RO$_2$ chemistry is the same except for the competition between RO$_2$ and OH. Also, any OH generated through photolysis would react within the photolysis reactor and would not be detected as a signal at long reaction times.

If the photolysis is stopped after say 20 shots, could the loss of ROOOH from the flowtube be monitored in some way?

Yes, we did such experiments. The signal was recorded for 20 shots before uncovering the photolysis laser, then the mixture was photolysed for 50 shots and then the laser was covered again for 50 shots: open black symbols are traces with photolysis laser covered and represent the average of all data points of that trace; full blue symbols represent traces with photolysis and have been obtained as the plateau from a mono exponential fit between 20 ms and the end of the trace. The dotted line is a guidance for the eye and shows the average of all data points for the first and last 20 shots.

The result for an isoprene experiment under conditions comparable to Figure 1 is shown in the following figure:

[Figure]

For the determination of the OH reactivity, was the same fitting window used for all of the decays, i.e. was it the same for the Pulse 1 decay as for the Pulse 40 decay? Or was a different start and end time used?

The same time window has been used for all traces: the full trace has been fitted, only the first 20 ms, showing some stray light from the photolysis laser as well as the rise of the OH signal, have been discarded.

There is no doubt that the work is interesting, and the idea that ROOOH decomposing to OH and hence constituting a potential interference for OH instruments is worthwhile, and needs further exploration. However, it is unknown what the mechanism of decomposition might be, and it is a complex chemical system and the model used to explain the laboratory results has many assumptions/simplifications – and there may be other species generated in the system which could release OH once inside the FAGE cell. There is likely to be a cascade of chemistry over the 40 photolysis shots generating many species (some of the products and intermediates formed will be different to the real atmosphere owing to the conditions and wavelength of light present).

We agree that we have not attempted to precisely understand what is going on in the photolysis volume. Again, one of the major pieces of evidence for our hypothesis is the fact that we see a clear difference in experiments with low and high VOC concentrations, i.e. under conditions where RO$_2$ react or not with OH radicals. Indeed, if the reaction of ROOOH with OH or its photolysis leads to rather stable species that in turns leads to OH signal in our FAGE, we would not be able to distinguish it. On the other hand, if something like this happens, it might also happen in remote environments.

The language in the paper needs moderating considerably regarding extrapolating the laboratory observations to the conclusion that the work presents convincing evidence that previous model-measurement uncertainties in low NOx environments containing isoprene is due to ROOOH (R from isoprene or similar) decomposition. Our knowledge of ROOOH abundance and removal processes is virtually non-existent, and the model has been optimised using adjustable parameters so as to give a level of OH in the instrument which becomes important compared with atmospheric levels. The wording over the implications of the work, and also the title of the paper, needs changing.

We have removed the word "convincing" from the manuscript and have also more often used the subjunctive form. We are fully aware that our work does not present a final proof that the disagreement between modeling and measurement can be resolved by taking into account ROOOH. Unfortunately,

we do not have any datasets with our FAGE system from remote environments that would allow us to test our hypothesis; this now could only be done by the corresponding groups that have these data sets.

We have also changed the title by adding an "a" and changing it to a question:

ROOOH: **a** missing piece of puzzle for OH measurements in low NO environments**?**

Other points:

Title, abstract and conclusions – these need to be toned down to report the observations made, that ROOOH needs to be considered as a potential source of OH inside instruments. Any implication that the puzzle is solved is going too far and is rather premature. The title cannot remain as is. The abstract and conclusion also need to make it clear that any OH from ROOOH would be highly dependent upon the instrument design, and make implicit any assumptions made in the model.

Words such as "convincing" are of course subjective – and should be toned down or removed completely.

The title has been changed (see above), we have removed "convincing", we have used several times the subjunctive form. We have also changed the abstract to the following:

*Abstract. Field campaigns have been carried out with the FAGE technique in remote biogenic environments in the last decade to quantify the in situ concentrations of OH, the main oxidant in the atmosphere. These data have revealed concentrations of OH radicals up to a factor of 10 higher than predicted by models, whereby the disagreement increases with decreasing NO concentration. This was interpreted as a major lack in our understanding of the chemistry of biogenic VOCs, particularly isoprene, which are dominant in remote pristine conditions. But interferences in these measurements of unknown origin have also been discovered for some FAGE instruments: using a pre-injector all ambient OH is removed by fast reaction before entering the FAGE cell, and any remaining OH signal can be attributed to an interference. This technique is now systematically used for FAGE measurements, allowing the reliable quantification of ambient OH concentrations along with the background OH. However, the disagreement between modelled and measured high OH concentrations of earlier field campaigns as well as the origin of the now-quantifiable background-OH is still not understood. We present in this paper the compelling idea that this interference, and thus the disagreement between model and measurement in earlier field campaigns, might be at least partially due to the unexpected decomposition of a new class of molecule, ROOOH, within the FAGE instruments. This idea is based on experiments, obtained with the FAGE set-up of University Lille, and supported by a modelling study. Even though the occurrence of this interference will be highly dependent on the design and measurement conditions of different FAGE instruments, including ROOOH in atmospheric chemistry models might reflect a missing piece of the puzzle in our understanding of OH in clean atmospheres.*

The balance of the main paper and the supplementary information seems skewed. There is information in the SI which really ought to be in the main paper (there is no compelling reason this has to be done for space reasons). e.g. Figures S1 and S4, S6 (this seems central to show) and probably one panel for butane and methane.

We have shifted a large part of the supplementary data into the main manuscript, notably the modelling of the chemistry in the photolysis reactor, all experiments and figures concerning the different tests with laser energy, different isoprene concentration and the tests with butane and $CH_4$.

Page 4, line 18, it should be 200 microseconds (not ms).

Has been corrected.

Figure 1. It is not clear how the t=0 maximum in OH signal varies with the number of shots? Does this change with pulse 1 to 40 in any systematic way?

The LIF intensity at time 0 decreases slightly, probably due to the slow depletion of $O_3$ within the photolysis volume. The following figure shows the LIF intensity at time 0 for the experiments from Figure 1 (note that the fit to Figure 1 has been changed compared to the initial manuscript after close inspection of the data with respect to this question, now taking only data from 20 msec on, while before it was 10 msec. This has a slight influence as well on the decay rate and the plateau, therefore Figure 2 has also slightly changed). From pure photolysis one would expect at each photolysis shot a decrease of the signal of : $OH / O_3 \approx 1.4 \times 10^{10} / 1.5 \times 10^{13} \approx 1 \times 10^{-3}$, in good agreement with the decrease of the OH signal at time 0 : $(1.8 \pm 1.5) \times 10^{-3}$.

[Figure]

Also what does "data analysis is carried out with raw data" mean?

We wanted to express that the averaging is just done for better visualisation: "raw data" has been replaced by "non-averaged" data.

Figure 2. The red line shows the plateau OH signal from the fit, gradually increasing with the number of photolysis pulses. Given the very small magnitude of the plateau OH compared with the initial OH signal the accuracy of the fit at long time is very important. Could Figure 1 at long times be plotted on a very expanded vertical scale (perhaps from y=0 to y=0.15) and the fits be shown with it? The results are very dependent on how well things fit at long times. Related to this point I would like to see Figure 2 plotted when the plateau is averaged using the data at later times. As the OH reactivity becomes smaller

with the number of photolysis pulses, it takes longer for the signal to reach the plateau, so does this slower decay promote the increasing baseline signal in any way? There are certainly some decays in the SI that do not seem to have reached the baseline before the averaging window to obtain the plateau signal begins.

We agree that the increase of the signal is very small compared to the initial signal, and also compared to the scatter and the uncertainty. However, we are confident that the results really show an increase in OH signal, because consistently we always observe this increase only when we are in conditions where $RO_2$ reacts with OH, at any other experiments the background is stable within the uncertainty.

We have now plotted the data from Figure 2 in the manuscript (former Figure 1) with an inset showing the pre-photolysis signal as well as the first and the last trace blown up vertically, together with their fits. It can be seen that even for the slowest decay (shot 40) the fit has reached the plateau value at around 0.3 sec. In the figure below we show the data from Figure 4 (former Figure 2, without error bars) together with the raw data averaged from 0.35 to 0.41 sec. While the scatter is much larger using the averaged values, the overall trend of an increasing background with increasing photolysis pulses is the same.

[Figure]

Page 6, lines 5-15. Excitation laser energy and photolysis energy are both used here – be clearer about which pulse energy was changed. Clearly Fig S2 and S3 are for probe (excitation) laser energies, but unclear if photolysis (266 nm) energy was changed also?

Everything else was kept constant, only the probe laser was varied. This has been clarified in the manuscript. This part has also been moved from the supplementary data to the main manuscript.

The error in [ROOOH] from the model after 40 shots (figure S6 model) needs to be stated.

The error in [ROOOH] is very large, the model just serves to get a rough estimate. As we state, there are many processes that are not taken into account in the model (diffusion into unphotolysed volume, wall loss, photolysis of ROOOH, reaction of OH with several products, inhomogeneous photolysis beam (this can make a large error for radical-radial reactions, but is very difficult to quantify) etc.). So we think that the uncertainty is at least a factor of 10, but rather over- than underestimated (most of the neglected processes would either consume or produce less ROOOH).

Why was $10^{-4}$ s$^{-1}$ chosen as the loss rate for ROOOH in the atmosphere? Presumably as this gave an OH concentration of around $1 \times 10^6$? There is a very large uncertainty in [ROOOH], and so the statement on page 9, line 8 that [ROOOH] is predicted to be of the order of 50-200 pptv seems very optimistic in terms of the range of concentrations? For Figure 4, were there RO$_2$ measurements made in the field to constrain the model, or was modelled RO$_2$ used?

We agree with the reviewer that there is very large uncertainty in the atmospheric abundance of ROOOH. As we state on page 4 line 15 of the manuscript, the removal rate (and dominant process) is unknown at present. The use of a loss rate of $10^{-4}$ s$^{-1}$ was chosen as an educated guess of a lifetime. This is on the order of the lifetime of ROOH (which is typically $> 1$ hour) and much longer than RO$_2$ (which is typically $< 10$ seconds). As we state on page 115 line 9 of the manuscript, we evaluated a range of 3 orders of magnitude in the loss rate of ROOOH in our modelling but for space reasons and to focus the discussion we opted to discuss the intermediate loss rate of $10^{-4}$ s$^{-1}$. We proposed the manuscript to make this clearer:

*As neither the removal rate nor the dominant process of these ROOOH species are currently known, different removal rates were tested, ranging from $10^{-5}$ to $10^{-2}$ s$^{-1}$.*

*Figure 3 shows the average diurnal peak concentration of ROOOH in the Boreal (left) and Austral (right) summer obtained using a removal rate of $10^{-4}$ s$^{-1}$, leading to ROOOH lifetimes of around 3 hours, on the same order as the lifetime of ROOH species. Peak concentrations of several 100 ppt are reached in this scenario, especially at tropical latitudes, which would lead to an interference in the UL-FAGE system of the order of $1 \times 10^6$ cm$^{-3}$. However, we would like to insist on the fact that both, the modelled concentration of ROOOH in the atmosphere as well as the sensitivity of the UL-FAGE against ROOOH species, bear currently an uncertainty of at least a factor of 10,*

For Figure 4 (Figure 10 in the revised manuscript) we did not constrain RO$_2$ but have made clearer in the text which species were constrained to the data reported in Feiner et al (2016)}

Discussion

Page 10, line 6.

It is stated that the product of the RO$_2$+OH reaction leads to an OH interference. The signal may originate from that, but there may be other origins of the signal also, and so the wording needs to be more flexible. This is also where a statement about the different FAGE designs is needed.

We think that the results of the different experiments carried out in this work show clearly that the increase in OH signal in the UL-FAGE does indeed originate from the products of the reaction of RO$_2$ + OH, so the wording seems suitable to us. There was already a (small) statement that other FAGE systems might behave differently (If occurring also with a comparable intensity in other FAGE instruments....), but we emphasized this fact by adding:

*The intensity or even the occurrence at all can depend on the design and working conditions of the FAGE set-up, which is different for different groups. However, if occurring also with a comparable intensity in other FAGE instruments, this interference might be high enough to explain numerous observations obtained with FAGE instruments from other research groups including: ......*

Supplementary material.

I think much of section 1 on the FAGE and probe system needs to go in the main paper.

We have added the Figure of the set-up into the main manuscript as well as a few details, but have preferred to leave most details of the set-up in the sup data.

What is the residence time of the FAGE detection cell (inlet from the pinhole and fluorescence cell)? There is no discussion of this and it is an important point. Was the pumping rate of the cell changed to change the residence time?

The volume of the FAGE cell between the inlet nozzle and the excitation laser beam is 0.25 l, the pumping into the FAGE is 3 l, this leads at 2 Torr to a residence time of the gas of around 1 sec. No, pumping rate was not changed in these experiments. Such experiment would indeed have been a good idea, but we did not do and now the experiment is in a different configuration.

Figure S2 – in the caption add a line saying what the black line is.

Has been done.

Line 187 of SI – "is consistent with" is better than "is indeed due to"

Has been done.

For all of the OH signal versus time plots the y axis ends at zero. Are there any negative points, or is the "baseline" some value a bit above zero?

There are never negative values, because the signal is obtained by counting photons. Therefore, it is either zero (if no photon occurred at that time window during any of the 20 individual decays) or above.

Figure S6. The concentrations versus no of photolysis pulses. This seems central and should be in the main paper.

Has been done

Line 220, reword "very basically"

Has been done.

---

## Author Response (AR2)

The author addressed the questions raised in the first review but one, the mechanism of ROOOH decomposing into OH.

As we state on page 14, we don't know the mechanism of the ROOOH leading to OH.

The original claim of the paper that ROOOH is the missing source of interference got toned down. Still I don't see how the night time increase of the interference reported by Mallik et al or Novelli et al can be explained by night time formation of ROOOH from RO2+OH. My last remaining minor issue is to ask the authors to note this contradicting published result in their work.

We have added twice (Page 14, line 21 and Page 17, line 28)
It is however not very likely that it can explain an increase in the interference at night, such as observed by (Novelli et al., 2014a).
However, we are not sure which measurements you mean by Mallik et al: the only paper we found by Mallik et al. is on measurements in Cyprus, and we could not find any information on OH interference at night (not even during the day).

By all my reservation that ROOOH is a major contributor to the LIF interference reported mainly by us and Bill Brune's group, the chemistry of RO2+OH and the product ROOOH is very interesting and therefore I recommend publication.

To address the remaining open question of the mechanism requires additional work that may be beyond the scope of this study. I still would like to encourage the authors to look into :
- to change the residence time after passing the pinhole to derive from the temporal behavior of OH loss and production terms inside as shown by Novelli et al.
- change the diameter and length of the inlet behind the pinhole to investigate if wall contact is important. If so, does the choice of material or coating at the wall makes a difference. For our LIF we tested and can exclude the walls being involved as we used Macrolon tm (plastic), stainless steel, aluminum anodized & not anodized and none changed the result.
- does the use of a glas capillary change the result as the gradual pressure drop would reduce the effect of a shock wave, indicating if the cluster formation is of any importance. For us I can neglect this.
- does a flat plate or just a conical beam skimmer vs a Laval nozzle as pin hole affect the outcome as the flow though a free expansion of the jet behind a flat plate still generates an eddy whereas this is surpassed in a Laval nozzle.
- does the use of a single beam vs multi pass cell makes a difference?

We agree that the ideas for additional measurements are interesting, but as you say, they are well beyond the scope of this paper. We would be much more motivated to do any more work on this subject, if we had data sets from remote biogenic environments, obtained with our FAGE and showing large disagreement with models. However, we do not have any such data. Therefore it seems useless to us to find out more information on how and why we observe OH in the presence of ROOOH.

---

## Author Response (AR3)

Christa FITTSCHEN
Christa.FITTSCHEN@univ-lille1.fr
+33 (0)3 20 33 72 66

Villeneuve d'Ascq, le 20 décembre 2018

To Frank KEUTSCH

Co-editor of

Atmospheric Chemistry and Physics

Dear Frank,

Please find enclosed a revised version of the manuscript entitled "ROOOH: a Missing Piece of Puzzle for OH Measurements in low NO Environments?" by Christa Fittschen, Mohamad Al Ajami, Sebastien Batut, Valerio Ferracci, Scott Archer-Nicholls, Alexander Archibald and Coralie Schoemaecker.

We have considered all your remarks in the new version, the changes with respect to the former version are marked in yellow.

We are looking forward to hearing from you.

Sincerely yours

Christa FITTSCHEN
Research Director, CNRS

Physico-Chimie des Processus de Combustion et de l'Atmosphère PC2A
UMR 8522 CNRS-Lille1    Plateforme de Métrologie Optique MeOL
Université Lille1 Sciences et Technologies
Cité scientifique – Bât C11/C5/CERLA - 59655 Villeneuve d'Ascq Cedex
Tél. +33 (0)3 20 43 49 31 | Fax. +33 (0)3 20 43 69 77
www.pc2a.univ-lille1.fr   www.meol.cnrs.fr